# Collaborative and Confidential Junction Trees for Hybrid Bayesian Networks

**Roberto Gheda**[*]
Delft University of Technology
r.gheda@tudelft.nl

**Abele Mălan**
University of Neuchâtel
abele.malan@unine.ch

**Thiago Guzella**
ASML The Netherlands B.V.
thiago.guzella@asml.com

**Carlo Lancia**
ASML The Netherlands B.V.
carlo.lancia@asml.com

**Robert Birke**
University of Turin
robert.birke@unito.it

**Lydia Y. Chen**
Delft University of Technology
y.chen-10@tudelft.nl

## Abstract

Bayesian Network models are a powerful tool to collaboratively optimize production processes in various manufacturing industries. When interacting, collaborating parties must preserve their business secrets by maintaining the confidentiality of their model structures and parameters. While most realistic industry scenarios involve hybrid settings, handling both discrete and continuous data, current state-of-the-art methods for collaborative and confidential inference only support discrete data and have high communication costs. In a centralized setting, Junction Trees enable efficient inference even in hybrid scenarios without discretizing continuous variables, but no extension for collaborative and confidential scenarios exists. To address this research gap, we introduce `Hybrid CCJT`, the first framework for confidential multiparty inference in hybrid domains with semi-honest, non-colluding adversaries, comprising: (i) a method to construct a strongly-rooted Junction Tree across collaborating parties through a novel construct of interface cliques; and, (ii) a protocol for confidential inference built upon multiparty computation primitives comprising a one-time alignment phase and a belief propagation system for combining the inference results across the Junction Tree cliques. Extensive evaluation on nine datasets shows that `Hybrid CCJT` improves the predictive accuracy of continuous target variables by 32% on average compared to the state-of-the-art, while reducing communication costs by a median $10.4\times$ under purely discrete scenarios.

## 1 Introduction

Bayesian Networks (BNs) are probabilistic models that incorporate domain knowledge while retaining interpretability [1]. This makes them a powerful tool for optimizing manufacturing processes in numerous industrial domains [2, 3, 4]. One example is semiconductor manufacturing, which involves complex procedures and machines. Production process optimization within the field increasingly requires the collaboration between parties like *manufacturers* (with domain knowledge on the product design and manufacturing steps) and *equipment vendors* (with expertise on the equipment performance) [5]. However, such collaboration must preserve the confidentiality of the BNs' structure and parameters, which encode sensitive industrial knowledge and the parties' evidence. Moreover, most real-world applications require models to handle discrete ($\Delta$) and continuous ($\Gamma$) data simultaneously [6], achieved using *Hybrid Bayesian Networks* [7].

---

[*]Work partly done while interning at ASML.

39th Conference on Neural Information Processing Systems (NeurIPS 2025).

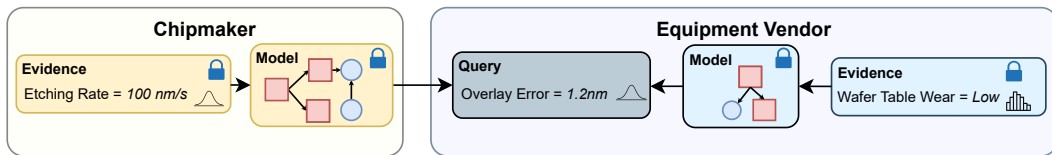

Figure 1: The equipment vendor and the chip manufacturer observe status of etching rate and wafer table wear separately (**evidences**), and model relationship between variables with BNs (**models**). Equipment vendor predicts the overlay error (**query**), leveraging evidence of wafer table wear (**discrete** value) and etching rate (**continuous** value).

*Collaborative Confidential Bayesian Networks* tackle the aforementioned challenges by treating local parties' BNs as a larger global network while concealing these parties' knowledge from each other. Figure 1 exemplifies confidential inference in the semiconductor industry.

Non-collaborative BNs can be queried via the *Variable Elimination* algorithm [7], which eliminates variables one by one by marginalization. CCBNet [8] extends this algorithm to collaborative confidential BNs. However, CCBNet has two main drawbacks. First, it only supports discrete variables, leaving out continuous ones. Second, it does not scale well, as inference time is worst-case exponential in the number of network nodes, and high communication costs to exchange partial results between parties. Such drawbacks hinder its real-world adoption in scenarios with continuous or mixed-variable types, and BNs with as few as 15 nodes.

To make inference on larger BNs tractable, [9] divides the network into smaller sub-networks, called *cliques*, and organizes them into a *Junction Tree* (JT) structure. Subsequently, inference is performed via the Variable Elimination algorithm in each clique, and the results are combined via *Belief Propagation* [10]. Conveniently, when Junction Trees are *strongly-rooted*, they can be used to carry-out inference over Hybrid BNs [11]. The strongly-rooted property requires a strong order eliminating continuous variables before discrete ones [12]. Constructing such a tree in a distributed and confidential fashion is an unexplored challenge.

To tackle this, we propose `Hybrid CCJT` (**Hybrid C**ollaborative **C**onfidential **J**unction **T**rees), the first peer-to-peer framework that runs collaborative inference over Hybrid BNs confidentially with no need for a central coordinator. `Hybrid CCJT` consists of two main components:

(i) A collaborative strongly-rooted Junction Tree. We introduce *interface cliques* in the tree structure, a type of clique that models distributions of variables common among parties, and preserves a strong order which is key for inference in hybrid BNs, detailed in Section 3.3.

(ii) A confidential inference protocol for Hybrid BNs. To perform inference, parties propagate their clique partial results over the built strongly-rooted JT, starting with the continuous variables and ending with the discrete ones. Within this protocol, we present two novel procedures to ensure confidentiality during inference when processing interface cliques: *Collaborative $\Delta$ Inference* and *Collaborative $\Gamma$ Inference*. In particular, the former eliminates more variables before message-passing than the state-of-the-art CCBNet for discrete BNs, drastically reducing communication costs. The latter enables handling continuous variables, detailed in Section 3.4.

Code is available at github.com/r-gheda/hybrid-ccjt. Our contributions are as follows:

- We propose `Hybrid CCJT`, the first framework for Collaborative Confidential Hybrid BN inference.

- We design a **collaborative strongly-rooted JT protocol**, the first method to build a collaborative, strongly-rooted Junction Tree based on interface cliques to run accurate inference in hybrid domains.

- We prove the correctness of this novel strongly-rooted JT structure and its construction protocol.

- We propose a **confidential inference protocol**, which uses HE and SMC to make parties' BNs collaborate, while protecting their structure and parameters of and posterior of private variables.

- We evaluate our method against nine different datasets and report improvements compared to non-hybrid confidentiality-preserving methods. We obtain a 32% average decrease in mean squared error and up to $86\times$ reduction in communication costs. Furthermore, our method uses up to $331\times$ smaller communication costs under purely discrete scenarios.

## 2 Related Studies and Background

### 2.1 Hybrid CLG Bayesian Networks

Bayesian Networks [1] are directed acyclic graphs whose nodes are random variables and whose edges correspond to the direct influence of one variable on another. The *conditional probability distribution* (CPD) of a variable $X$ given its parents, $P[X \mid \mathrm{pa}(X)]$, is called its *factor*. The tables that summarize such probability distributions are called CPD tables. Traditionally, BNs only allow variables to be discretely valued [7]. However, such a requirement limits the representation quality for variables that naturally describe continuous-valued data [13]. Moreover, exact inference in discrete BNs is NP-hard, while other continuous representations, such as *Conditional Linear Gaussian* (CLG), can perform exact inference with polynomial cost in the network size [7].

Hybrid Bayesian Networks [13] allow to model probability distributions with both discrete ($\Delta$) and continuous variables ($\Gamma$). One of the most common classes of hybrid models is the set of *Hybrid Conditional Linear Gaussian* (Hybrid CLG) distribution models [14]. In this class, continuous variables are Gaussian-shaped and cannot have any discrete children. The factor of a continuous variable $X \in \Gamma$ with discrete parents $z_\Delta$ and continuous parents $z_\Gamma$ is given by:

$$P[X|\{z_\Delta, z_\Gamma\}] = \mathcal{N}(X; \alpha(z_\Delta) + \beta(z_\Delta)^T z_\Gamma, \sigma^2(z_\Delta)) \tag{1}$$

where $\alpha$ and $\beta$ are the coefficients that depend on the discrete state combination of $z_\Delta$. If the state combination of $z_\Delta$ is fixed, $X$ is Gaussian-shaped. Otherwise, $f(X)$ is a mixture of $\mathcal{O}(2^{|\Delta|})$ Gaussian distributions. In general, even representing the correct marginal distribution in a Hybrid CLG network requires space exponential in the size of the network [7]. Furthermore, even approximate inference for simple model structures such as polytrees is NP-hard in Hybrid CLG networks [15].

**Inference** Lauritzen and Jensen [11] propose an algorithm to carry out accurate inference in Hybrid CLG BNs by leveraging a strong elimination order. In such an order, continuous nodes get eliminated from the graph before discrete ones [12].

### 2.2 Collaborative and Confidential Bayesian Networks

Collaborative confidential BNs aim to hide each party's own BN structures and parameters from one another during distributed inference. CCBNet [8] is the current state-of-the-art for collaborative confidential BN inference. It is based on two protocols: CABN (Confidentially Augmented Bayesian Networks) and SAVE (Share Aggregation Variable Elimination). CABN privately performs *alignment* of factors of common variables, while SAVE performs distributed inference based on a Variable Elimination and BN merging scheme inspired by Del Sagrado and Moral [16] and Feng et al. [17].

Despite being a significant step forward in the Confidential BN literature, CCBNet falls short in several aspects. It only supports discrete variables, has high communication costs, and reveals marginals for some private variables from peers to the party executing a query.

**Multiply-sectioned Bayesian Networks** [18] proposed a method for distributed inference on Bayesian Networks leveraging Belief Propagation. However, these methodologies require analyzing the shared BN structure, breaking confidentiality requirements. Furthermore, loopy belief propagation entails sending more messages (due to the iterative nature of the algorithm) and is proven to converge only on graphs with at most one cycle [19].

## 3 Hybrid CCJT

### 3.1 Preliminaries on Junction Trees

Exact inference in discrete BNs is NP-hard as its cost grows exponentially in the network's number of variables (nodes). A widely used technique to make inference on larger instances tractable is to build a Junction Tree [9]. Similarly to tree decomposition, the original network is decomposed into a tree-like graph where each node contains a sub-graph of the original BN—called *clique*. After computing inference in each small clique (e.g., by variable elimination), *Belief Propagation* [7], a message-passing algorithm proven to converge in linear time on trees, can combine the results. As such, the size of the largest clique bounds the cost of inference.

In Figure 2a we showcase the popular ASIA network with 8 nodes [9] as an example. Here, running Variable Elimination requires $\approx 2^8$ operations to perform exact inference. In contrast, on the corresponding Junction Tree in Figure 2a, the size of the largest clique (4 in our example) bounds the inference cost. Thus, it requires only $\approx 2^4$ operations.

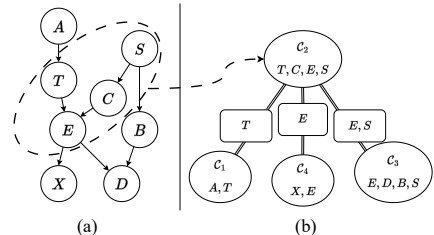

Formally, a Junction Tree of a Bayesian Network over variables $\mathcal{X}$ with set of factors $\Phi$ is a computational graph whose nodes $C_i$, termed *cliques*, are tuples $(X_i \subseteq \mathcal{X}, \Phi_i \subseteq \Phi)$. Edges, termed *separators*, are associated with a set of variables called *sepset* $S_{i,j} = X_i \cap X_j$. These edges connect the cliques to form a tree. To be valid, a Junction Tree must satisfy the following rules:

Figure 2: Construction of a Junction Tree: (a) Example of a discrete Bayesian Network. (b) A corresponding Junction Tree consisting of 4 cliques.

- *Family preservation*: Each factor $\phi \in \Phi$ must be associated with one clique $C_i$ such that $\text{Scope}[\phi] \subseteq X_i$.
- *Running-intersection property*: For each pair $C_i$ and $C_j$, every clique on the path between them contains $X_i \cap X_j$.

Exact inference can run on Junction Trees via message passing schemes exchanging factors with joint probabilities, one of the most popular being the sum-product algorithm [10]. Let us define the potential of each clique $C_i$ as $\phi(C_i) = \prod_{\phi_j \in \Phi_i} \phi_j$. Sum-product requires cliques to send messages through the tree towards a root as follows:

$$\mu_{C_i \rightarrow C_j} = \sum_{v \notin S_{i,j}} \phi(C_i) \prod_{C_k \in \text{nb}(C_i) \backslash C_j} \mu_{C_k \rightarrow C_i} \tag{2}$$

where $\mu_{C_k \rightarrow C_i}$ is a message sent from clique $C_k$ to clique $C_i$ in the form of a CPD table with scope $S_{k,i}$. $\sum$ represents the variable *marginalization* operator: a fundamental operator of BN inference which removes a set of variables from a factor. The outcome of such an operation is called *marginal*.

**Strongly-Rooted Junction Trees** Regular Junction Trees lack a strong elimination order [12]. They break the requirement of strictly eliminating continuous nodes before discrete nodes for correctly running inference on Hybrid CLG models.

**Definition 3.1.** A Junction Tree over a set of discrete variables $\Delta$ and CLG variables $\Gamma$ is strongly-rooted if it has a distinguished clique $C_r$, called *strong root*, such that for every couple of neighboring cliques $(C_i, C_j)$, with $C_i$ being closer to $C_r$ than $C_j$, it holds that:

$$C_i \cap C_j \subseteq \Delta \quad \vee \quad C_j \backslash C_i \subseteq \Gamma \tag{3}$$

Lauritzen and Jensen [11] shows that strongly-rooted Junction Trees allow computing exact posteriors of all discrete variables (*strong marginalization*) and exact first and second moments of all continuous variables (*weak marginalization*) via multivariate Gaussian approximation.

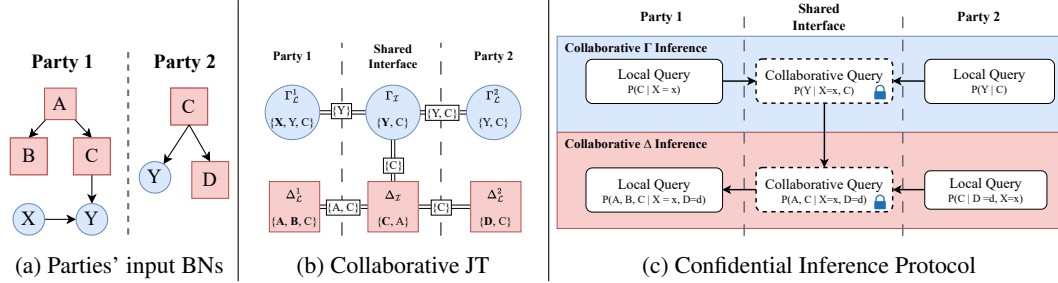

Figure 3: Overview of `Hybrid CCJT`. Red represents discrete, blue represent continuous. (a) Parties' secret BNs are taken as input. (b) A Junction Tree is constructed. (c) Confidential inference is performed. Arrows represent input.

### 3.2 `Hybrid CCJT` **Overview**

`Hybrid CCJT` is a framework to run collaborative, confidential queries in Hybrid CLG domains. It uses two main protocols: (i) **Collaborative strongly-rooted JT Protocol** for parties to collaboratively generate a strongly-rooted Junction Tree; and (ii) **Collaborative Inference Protocol** for parties to jointly perform exact inference leveraging homomorphic encryption (HE) [20] and secure multi-party computation (SMC) [21] to preserve confidentiality. Figure 3 showcases the steps of `Hybrid CCJT`. Further, Appendix H provides a detailed diagram that explains the steps of inference in a two parties scenario.

We assume that variables from different parties have the same name only if they represent the same concept, and that considering the distinct parents for the same node across parties to be independent reasonably approximates the ground truth. We model the joint probability distribution as the union of each party's BN. Our **adversarial model** includes **semi-honest** parties that follow the protocol while trying to abuse gained information [22] but do not collude. No trusted third party is assumed. The goal is to protect the structure and parameters of parties' BNs, and to hide marginals and posteriors of private variables. We prove our confidentiality achievements in Appendix B.

In centralized settings, **Junction Tree construction** requires using algorithms that analyze the network structure [7]. Doing so goes against our confidentiality requirements and does not consider minimizing communication costs. Our collaborative strongly-rooted JT Protocol allows the construction of a strongly-rooted Junction Tree without sharing such confidential information. In this regard, two key novel steps are the construction of interface cliques across parties and modeling common variable distributions. Centralized **inference** requires parties to reveal the network structure and marginals within the interface clique. Our Confidential Inference Protocol allows computing the outcome of this inference without disclosing such sensitive information.

### 3.3 Collaborative Strongly-Rooted JT Protocol

In centralized settings, building Junction Trees involves algorithms that analyze the full network structure [7]. Such an approach goes against our confidentiality requirements and does not account for the demand to reduce communication costs. Contrarily, our Collaborative Junction Tree protocol builds a strongly-rooted Junction Tree without sharing confidential information. First, we construct tree cliques and separators to define an elimination order valid for hybrid inference. We then model the posterior distribution of common variables by a process we call *alignment*. The assumptions we make for providing a correct alignment are as follows: (i) variables have the same name when defining identical concepts; (ii) same variables have the same discrete states or continuous measurement units.

Every party defines an input BN (either Discrete, Continuous, or Hybrid CLG) describing their knowledge (Figure 3a). Network structure and parameters can be defined by human experts or deduced via structure and parameter learning methods [23]. Before running the protocol, we find the set of *interface* variables $\mathcal{I}$ (i.e., the common variables among parties). For confidentiality, we do so via *Private Set Intersection* [24]. The Junction Tree's definition relies on each party's input network and the global set of interface variables.

**Junction Tree construction** We construct the collaborative Junction Tree by building each party's local discrete and continuous cliques and their corresponding shared interface cliques. Finally, we connect all the cliques. For each party $i$, we define two *local cliques*: one discrete ($\Delta_{\mathcal{L}}^i$) and one continuous ($\Gamma_{\mathcal{L}}^i$) (see red and blue nodes in Figure 3b). Then, we define two shared *interface cliques*, one discrete ($\Delta_{\mathcal{I}}$) and one continuous ($\Gamma_{\mathcal{I}}$). Next, we map variables and factors to each clique. Let $\Phi(X)$ denote the set of factors for each $x \in X$, where $X$ is a set of variables. Take $\Delta_i$ and $\Gamma_i$ as the set of discrete and continuous variables in party $i$'s input network. Let $\text{Scope}_i[x]$ be the scope of variable $x$ in party $i$'s input network. Each party $i$ defines its local cliques as $\Delta_{\mathcal{L}}^i = \left(\Delta_i, \Phi(\Delta_i \setminus \mathcal{I})\right)$ and $\Gamma_{\mathcal{L}}^i = \left(\Gamma_i \cup \text{Scope}_i[\Gamma_i], \Phi(\Gamma_i \setminus \mathcal{I})\right)$. Define interface cliques as $\Delta_{\mathcal{I}} = \left(\text{Scope}[\bigcap_i \Delta_i] \cup (\text{Scope}[\Gamma] \cap \Delta), \Phi(\Delta \cap \mathcal{I})\right)$ and $\Gamma_{\mathcal{I}} = \left(\text{Scope}[\bigcap_i \Gamma_i] \cup (\text{Scope}[\Gamma] \cap \Delta), \Phi(\Gamma \cap \mathcal{I})\right)$.

Cliques are then connected to form a tree. As shown in Figure 3b, discrete and continuous local cliques are connected with discrete and continuous shared interface cliques, respectively. The two interface cliques are connected, with the separator being the set of *threshold variables* $\mathcal{T}$. That is, the set of discrete variables with continuous children ($\Delta \cap \text{pa}(\Gamma)$). The resulting graph is a valid Junction Tree, with any $\Delta_{\mathcal{L}}^i$ being a valid strong root. We show this in detail in Appendix A.

**Algorithm 1** Collaborative Inference Protocol

---

**Input:** discrete target $T^\Delta$, continuous target $T^\Gamma$, evidence $\mathcal{E}$, querier $Q$, peers $P$, overlap variables $O$

1: *CollaborativeContinuousInference*$(\mathcal{E}^\Gamma, \{Q\} \cup P, O)$
2: strong_marginals $\leftarrow$ *CollaborativeDiscreteInference*$(\mathcal{E}^\Delta, Q, P, O)$
3: $\Delta$-result $\leftarrow \sum_{var \in \{\Delta \setminus T^\Delta\}}$ strong_marginals     ▷ Local Variable Elimination
4: $\Gamma$-result $\leftarrow$ *WeakMarginalization*(strong_marginals, $T^\Gamma$)     ▷ Equation 10
5: **return** $\Delta$-result, $\Gamma$-result

---

**Interface alignment** After finding consensus on how to model probability distributions of common variables in the interface clique, parties can perform all further collaborative inference queries.

To align the CPD of discrete interface clique, we use a geometric average (also known as the *Logarithmic Opinion Pool* [25]):

$$P[A = a_j] = \frac{\prod_{i \in \mathbb{P}} P_i[A = a_j]^{w_i}}{\sum_{k \in \Omega(A)} \prod_{i \in \mathbb{P}} P_i[A = a_k]^{w_i}} \propto \alpha \prod_{i \in \mathbb{P}} P_i[A = a_j]^{w_i} \tag{4}$$

where $w_i$ is a publicly known weight of party $i$, which represents the confidence in its BN. Other foundational works on combining BNs [16] and consensus belief [25] also study this approach. From Equation 4, the definition of the interface clique potential for discrete variables follows:

$$\phi(\Delta_\mathcal{I}) = \alpha \prod_{i \in \mathbb{P}} \phi(\Delta_\mathcal{I}^i)^{w_i} \tag{5}$$

where $\alpha$ is the column normalization factor applied to each table's column based on the alignment process. To implement it confidentially, parties allocate space in the interface variables' CPDs to account for parents managed by other parties with obfuscated names. Then, they collaboratively compute column normalization factors $\alpha$ via HE [20]. This normalization is only applied during inference by the querying party, enabling peers to marginalize their private variables before message passing, where CPD entries are secretly shared via a multiplication-based scheme [26], enhancing both communication costs and privacy guarantees.

To align the continuous interface clique, we define the joint probability distribution of common continuous variables as the weighted mixture of parties' (local) Gaussian distributions. Given a fixed discrete state combination $s \in \Omega(\mathcal{T})$:

$$P[X | \mathcal{T} = s] = \sum_{i \in \mathbb{P}} w_i \mathcal{N}(X; \mu_{i,s}(X), \sigma_{i,s}(X)) \tag{6}$$

At inference time, we approximate this distribution as a normal distribution with the same first and second moments of Equation 6:

$$P[X | \mathcal{T} = s] = \mathcal{N}\Big(X; \sum_{i \in \mathbb{P}} w_i \mu_{i,s}(X), \sum_{i \in \mathbb{P}} [w_i \sigma_{i,s}(X)] + \sum_{i \in \mathbb{P}} [w_i(\boldsymbol{\mu} - \mu_{i,s}(X))^2]\Big) \tag{7}$$

where $\boldsymbol{\mu} = \sum_{i \in \mathbb{P}} w_i \mu_{i,s}(X)$ is the mean of the PDF. The abovementioned step does not require sharing any parameters, maintaining confidentiality.

### 3.4 Collaborative Inference Protocol

We outline the protocol via pseudocode in Algorithm 1 and graphically in Figure 3c. Let $Q$ be the querying party. Belief propagates towards $\Delta_\mathcal{L}^Q$ through two steps. First, parties collaboratively run inference within the continuous domain to compute marginals over threshold variables (line 1). Second, they run collaborative discrete inference (line 2). Subsequently, party $Q$ uses the computed marginals to perform the remaining part of the query (lines 3-4). Centralized inference requires parties to reveal the network structure and partial inference results inside of the interface clique. Our protocol provides a procedure to compute the outcome of this inference without disclosing such sensitive information. We discuss this in detail in Appendix B.

**Collaborative Continuous Inference** During this step, we aim to merge parties' local continuous evidence to find the strong marginals over threshold variables. Algorithm 2 outlines its logic.

---
**Algorithm 2** Collaborative Continuous Inference

**Input:** evidence $\mathcal{E}$, parties $P$, overlap variables $O$

1: **for** $p \in P$ **do**
2:     $p.\text{ovBelief} \leftarrow P[O \mid \mathcal{E}]$                                   ▷ Local Continuous Inference
3: **end for**
4: $\text{ovBelief} \leftarrow secretShare(\bigcup_{p \in P} p.\text{ovBelief})$                     ▷ Equation 8
5: **for** $p \in P$ **do**
6:     $\text{threshold} \leftarrow p.\Delta\text{-vars} \cap \text{pa}(p.\Gamma\text{-vars})$
7:     $p.\text{threshBelief} \leftarrow P[\text{threshold} \mid \text{ovBelief}]$
8: **end for**

---

**Algorithm 3** Collaborative Discrete Inference

**Input:** evidence $\mathcal{E}$, querier $Q$, peers $P$, overlap variables $O$

1: **for** $p \in P$ **do**
2:     $p.\text{factors} \leftarrow p.\text{threshBelief} \prod_{X \in \Delta} P[X \mid \text{pa}(X), \mathcal{E}]$
3:     $p.\text{msg} \leftarrow \sum_{var \in \{\Delta \setminus O\}} p.\text{factors}$                     ▷ Local Discrete Inference
4: **end for**
5: $Q.\text{ovBelief} \leftarrow \text{secretShare}(\{p.\text{msg} \ \forall p \in P\})$               ▷ Equation 9
6: **return** $Q.\text{ovBelief} \prod_{X \in \Delta} P[X \mid \text{pa}(X), \mathcal{E}]$

---

Messages from $\Gamma_{\mathcal{L}}^i$ to $\Gamma_{\mathcal{I}}$ are derived by each party without interaction (lines 1-3). When computing a message from $\Gamma_{\mathcal{I}}$ to $\Delta_{\mathcal{I}}$, we aim to find strong marginals over discrete parents of continuous variables. To do so, we merge the knowledge of continuous interface variables (line 4). Parties then marginalize all continuous variables to find strong marginals (lines 5–7). To merge continuous variable knowledge, parties collaboratively compute the mean and variance of Equation 7:

$$\boldsymbol{\mu}(X) = \sum_{i \in \mathbb{P}} w_i \mu_i(X), \quad \boldsymbol{\sigma}(X) = \sum_{i \in \mathbb{P}} [w_i \sigma_{i,s}(X)] + \sum_{i \in \mathbb{P}} [w_i(\boldsymbol{\mu} - \mu_{i,s}(X))^2] \tag{8}$$

We use additive secret-sharing with no trusted third party to preserve confidentiality [27] (line 4). Each party randomly splits its secret value into as many shares as the number of parties and sends them to each of them. Every party adds up the values it received. Finally, parties share their results and compute the sum to find the final output. To compute $\boldsymbol{\mu}$, each party $i$ secretly shares $w_i \mu_i(X)$. While to find $\boldsymbol{\sigma}$, each party $i$ secretly shares $w_i \sigma_i(X) + w_i(\boldsymbol{\mu} - \mu_i(X))^2$.

Finding strong marginals requires integrating out all continuous variables while accounting for their evidences. This can be conveniently done in $\mathcal{O}(|\Gamma|^3)$ using canonical form representation [7], which we discuss in detail in Appendix C.

**Collaborative Discrete Inference** After computing the marginals of threshold variables, parties finalize strong marginalization by collaboratively calculating the message from $\Delta_{\mathcal{I}}$ to $\Delta_{\mathcal{L}}^Q$. Algorithm 3 showcases the procedure. Each non-querying party $i$ finds posteriors over their discrete domain $\Delta_i$. Then, from Equation 2 and 5, we derive:

$$\mu_{\Delta_{\mathcal{I}} \to \Delta_{\mathcal{L}}^Q} = \alpha \phi(\Delta_{\mathcal{I}}^Q)^{w_Q} \prod_{i \in \mathbb{P} \setminus \{Q\}} \sum_{x \notin \Delta_Q} \phi(\Delta_{\mathcal{I}}^i)^{w_i} \mu_{\Delta_{\mathcal{L}}^i \to \Delta_{\mathcal{I}}} \tag{9}$$

We show the derivation of Equation 9 in Appendix E. From the equation follows that each non-querying party $i$ has to compute message $\text{m}_i = \sum_{x \notin \Delta_Q} \phi(\Delta_{\mathcal{I}}^i)^{w_i} \mu_{\Delta_{\mathcal{L}}^i \to \Delta_{\mathcal{I}}}$. It is possible to do so locally, as $\phi(\Delta_{\mathcal{I}}^i)^{w_i}$ is the outcome of the discrete interface alignment procedure known to party $i$ and $\mu_{\Delta_{\mathcal{L}}^i \to \Delta_{\mathcal{I}}} = \sum_{x \notin \mathcal{I}} \phi(\Delta_{\mathcal{L}}^i)$ can be computed without interaction between parties (lines 2-7).

Eventually, the product of $\text{m}_i$ gives a factor over variables owned by the querying party $Q$ that does not reveal any information about the posterior of variables owned by other parties. We prove the above statement in subsection B.2. To protect the content of these messages, we use a secret sharing scheme for multiplication [26]. A secret value gets split into shares distributed amongst parties. Parties perform the computation with their local share of each secret, and all aggregate their results to reconstruct the answer.

Table 1: Results on hybrid data: best in bold, second best Brier score underlined. Lower is better.

| Dataset #Parties Overlap | | | Healthcare | | | | Sangiovese | | | |
|---|---|---|---|---|---|---|---|---|---|---|
| | | | 2 | | 4 | | 2 | | 4 | |
| | | | 10% | 30% | 10% | 30% | 10% | 30% | 10% | 30% |
| **Hybrid CCJT** | | Brier ↓ | 0.0496 | **0.036** | **0.0577** | **0.0856** | 0.019 | **0.0129** | 0.02746 | 0.015 |
| | | MSE ↓ | **4.7e+06** | **4.6e+05** | **4.8e+06** | **1.4e+07** | 0.0033 | 0.0018 | **0.00041** | **0.0045** |
| | | Comm. ↓ | 4.7 | 16.6 | 11.4 | 139.7 | **43.5** | **87** | 219 | **596** |
| **Δ-CCJT** | 3 States | Brier ↓ | 0.0557 | 0.058 | 0.0651 | 0.128 | 0.0457 | 0.0138 | 0.0484 | **0.0125** |
| | | MSE ↓ | 5.9e+06 | 4.9+05 | 5.4e+06 | 1.7e+07 | 0.044 | 0.021 | 0.083 | 0.0071 |
| | | Comm. ↓ | 4.6 | **9.3** | 8.3 | **23.9** | 44.6 | 164.8 | **133.6** | 2654 |
| | 5 States | Brier | 0.0502 | 0.0578 | 0.0623 | 0.173 | 0.0243 | 0.0132 | 0.0476 | 0.013 |
| | | MSE ↓ | 5e+06 | 5.2e+05 | 5.3e+06 | 1.6e+07 | 0.025 | 0.012 | 0.012 | 0.0218 |
| | | Comm. ↓ | 4.6 | 18.9 | **5.2** | 36.2 | 71 | 261.3 | 216.3 | 4174 |
| | 10 States | Brier ↓ | **0.0488** | 0.044 | 0.0597 | 0.112 | **0.0181** | **0.0129** | 0.0269 | 0.013 |
| | | MSE ↓ | 4.8e+06 | 4.9e+05 | 5e+06 | 1.6e+07 | 0.012 | 0.0036 | 0.0069 | 0.0049 |
| | | Comm. ↓ | **4.0** | 58.2 | **5.2** | 66.7 | 140.1 | 213.7 | 424.7 | 24013 |
| **CCBNet** | 3 States | Brier ↓ | 0.0558 | 0.0568 | 0.0651 | 0.128 | 0.0496 | 0.0138 | 0.05769 | 0.0125 |
| | | MSE ↓ | 5.8e+06 | 4.9+05 | 5.4e+06 | 1.7e+07 | 0.044 | 0.02 | 0.064 | 0.007 |
| | | Comm. ↓ | 38.7 | 15.2 | 17.0 | 34.3 | 196 | 793.6 | 19044 | 20271 |
| | 5 States | Brier ↓ | 0.0502 | 0.0571 | 0.0623 | 0.172 | 0.0243 | 0.0132 | 0.0464 | 0.013 |
| | | MSE ↓ | 5e+06 | 5.2e+05 | 5.3e+06 | 1.6e+07 | 0.025 | 0.011 | 0.011 | 0.0215 |
| | | Comm. ↓ | 12.6 | 28.2 | 9.6 | 161.1 | 52.8 | 808.9 | 4273.3 | 377341 |
| | 10 States | Brier ↓ | **0.0488** | 0.044 | 0.0597 | 0.113 | **0.0181** | **0.0129** | 0.0269 | 0.013 |
| | | MSE ↓ | 4.8e+06 | 4.9e+05 | 5e+06 | 1.6e+07 | 0.012 | 0.0037 | 0.0069 | 0.0049 |
| | | Comm. ↓ | 9.7 | 135.2 | 9.6 | 560.2 | 154.1 | 1381.9 | 4074.6 | 189650 |

**Weak Marginalization** After computing discrete posteriors, following the weak marginalization procedure [7] yields the continuous ones. Thus, for any continuous variable $X$ we get:

$$P[X] = \mathcal{N}\Big(X; \sum_{s \in \Omega(\Delta_Q)} P[\mathcal{T}_Q = s]\mu_s(X), \sum_{s \in \Omega(\Delta_Q)} P[\mathcal{T}_Q = s]\{\sigma_s(X) + (\boldsymbol{\mu} - \mu_s(X))^2\}\Big) \quad (10)$$

where $\boldsymbol{\mu} = \sum_{s \in \Omega(\mathcal{T}_Q)} P[\mathcal{T}_Q = s]\mu_s(X)$ is the mean of the PDF in Equation 10.

**Proof of Confidentiality** Below, we outline why both of `Hybrid CCJT`'s collaborative phases preserve confidentiality during inference:

1. **Collaborative Discrete Inference** CPD tables of interface variables (say, $X$) are "augmented" to allocate space for all parents of $X$ pa$(X)$, some of which might be private. This step is required to ensure the correct outcome of HE, CPD product, and CPD normalization as in Equation 4. We show that this step does not leak any information about such private parent variables by proving that their marginals yield a uniform distribution $\mathcal{U}(\Omega(\text{pa}(X)))$. Where $\Omega(X)$ is the set of states of variable $X$. We prove this in Proposition B.2. Then, we prove that, after message passing, the marginal over these parent variables remains uniform. This ensures no information about such private variables leaks at inference time. We prove this in Proposition B.5. Furthermore, note that parties encrypt the names of variables and their states to enhance confidentiality further.

2. **Collaborative Continuous Inference** Unlike discrete variables, continuous counterparts do not require computing and correctly applying normalization factors potentially defined over private factors. Thus, continuous private variables are inherently protected during alignment. At inference time, the means and variances of shared continuous variables are calculated using multi-party secret sharing, which updates the parameters of such shared continuous interface variables. This approach inherently safeguards the information of private variables, as they are not involved in the update process.

## 4 Evaluation

We evaluate `Hybrid CCJT` on nine publicly available models (see Appendix G for details) whose data structures are hybrid or discrete only. We compare it against the state-of-the-art on different types of queries. Appendix F presents additional results on purely continuous data. Appendix I and J present further experiments on computational costs of cryptographic tools and insights on communication costs respectively.

**Evaluation Metrics** Our experiments assess the average predictive performance for discrete and continuous target variables, and the associated communication costs. For discrete variables, we evaluate the prediction quality using the Brier Score defined as $\frac{1}{N} \sum_{t=1}^{N} \sum_{i=1}^{R} (f_{it} - o_{it})^2$ where $N$ is the number of queries, $R$ is the number of target variables state combinations and $f$ and $o$ are the predicted and actual probabilities, respectively. We use the Mean Squared Error (MSE) of predicted means compared to the actual values for continuous variables. The average CPD and CLG parameter values transmitted per query give the communication costs.

**Dataset** We consider the models listed in Appendix G. We sample a dataset from each model. Then, we assign a subset of variables to each party. Each party receives the vertical split of the sampled dataset corresponding to its assigned variables. From these vertical splits, the parties independently learn the input BNs. The network structures are learned via 2-phase Restricted Maximization [28]. Parameters are learned via Maximum Likelihood Estimator for conditional probabilities (for discrete variables) and least squares regression models (for CLG variables) [29].

**Baselines** We compare `Hybrid CCJT`'s performance against two baselines:

- **CCBNet** [8]: the current state-of-the-art for collaborative confidential BNs.
- **Δ-CCJT**: a simplified `Hybrid CCJT` with only the discrete inference from Algorithm 3.

Since none of the baselines can handle continuous data, we discretize such variables with different degrees of coarseness, ranging from 3 to 10 states per variable. Given the complexity of discrete exact inference [7], a finer discretization implies significantly higher computational and communication costs, making it infeasible to run these algorithms with many states per variable.

### 4.1 Results on Hybrid Data

We consider three hybrid datasets: Healthcare and Sangiovese (Table 1), followed by Mehra (Table 2). We test `Hybrid CCJT` with different combinations for the number of involved parties and overlap ratios. The overlap ratio denotes the fraction of variables assigned to more than one party. For each, we run 1000 queries with one discrete target variable and 1000 with one continuous target variable.

Table 2: `Hybrid CCJT` results on the large hybrid dataset Mehra for 8 parties.

| **Overlap** | 10% | 30% |
|---|---|---|
| Brier | 0.00783 | 0.00772 |
| MSE | 7.4e+11 | 4.5e+12 |
| Comm | 186 | 6734 |

**Predictive Accuracy** In all experiments, `Hybrid CCJT` outperforms all baselines in predictive accuracy of continuous target variables with an average 32% improvement in MSE compared with the best-performing baseline. When targeting discrete variables, `Hybrid CCJT` is either the best performing solution or the second best performing solution with a performance gap always under $10^{-3}$ in terms of Brier score. The only exception is Sangiovese with 4 parties and 30% overlap, where the deficit to the best model is 0.0025 (0.0125 versus 0.015). Note that Sangiovese has only one discrete variable, and this column returns a quasi-uniform posterior distribution regardless of the set of continuous evidence. The aforementioned explains why running hybrid inference on this data does not lead to any improvement over column discretization. As one would expect, the best performing baseline is the one with a finer discretization. Using a coarse representation leads to a drastic performance decay compared to our implementation with an MSE 26.9 times higher, and a Brier score $42.8\%$ higher on average. In summary, `Hybrid CCJT` brings notable improvements when targeting continuous variables, proving the benefit of natively handling continuous data. Besides, `Hybrid CCJT` matches the baselines when targeting discrete variables.

**Communication Costs** `Hybrid CCJT` demonstrates superior scalability in communication costs compared to all discretized baselines. Although Δ-CCJT achieves the lowest communication costs on the smaller Healthcare dataset, this advantage diminishes with larger datasets where we can start to appreciate the improved scalability of `Hybrid CCJT`. Under Sangiovese with all continuous variables except one, the communication cost of Δ-CCJT with 10 states increases significantly faster than for `Hybrid CCJT`, reaching up to 40 times more communicated values per query. This happens because discrete CPD tables take more space than regular continuous posteriors. Further highlighting the advantage of handling continuous data natively. While discretizing with fewer states may reduce communication costs in specific scenarios, this comes at the expense of a sharp decline in predictive performance. For example, on Sangiovese with 4 parties and 10% overlap, Δ-CCJT exhibits nearly $2\times$ the error for discrete targets and $200\times$ the error for continuous ones, making

`Hybrid CCJT` the most desirable choice overall. The current state-of-the-art, CCBNet, exhibits the worst communication performance across all experiments. Its worst result averages almost 190K communicated values per query against only 596 used by `Hybrid CCJT` (i.e., a $318\times$ reduction). This shows how our collaborative discrete inference approach alone significantly improves communication costs. We explore this further in subsection 4.2.

**Large dataset** Mehra is the largest dataset we consider, with 4 times the number of parameters of Munin, the largest discrete dataset. While `Hybrid CCJT` managed to complete all experiments (see Table 2), none of the discretized baselines finished within the timeout[2].This is due to the heavy computational requirements of aligning large CPD tables of discretized continuous variables. Despite the size, `Hybrid CCJT` achieves good Brier and MSE scores, while maintaining reasonable communication costs. Specifically, for a 30% overlap ratio in the smaller Sangiovese dataset, `Hybrid CCJT` communicates less than a third of the values compared to $\Delta$-CCJT with 10 states.

## 4.2   Results on Discrete Data

Since scalability of communication costs is a significant issue for CCBNet, we emphasize the improvement of `Hybrid CCJT` on six models with purely discrete data. Table 3 showcases the results. On the Child, Alarm, and Insurance datasets, we run experiments with 2 to 8 involved parties, and up to 128 on larger datasets. We perform 2000 different queries for each. Since these datasets lack any continuous variable, `Hybrid CCJT` degenerates into $\Delta$-CCJT. In smaller experiments, with two parties, `Hybrid CCJT` reduces CCBNet communication costs by 8 times, from an average of 125 to 15 communicated values. Improvement factors further increase when the number of involved parties grows. In larger-scale experiments, CCBNet's communication costs increase

Table 3: Results on discrete data: communication costs on discrete datasets. 10% overlap. Lower is better for all. Predictive accuracy difference is negligible ($< 10^{-3}$).

| Dataset | #Parties | CCBNet | Hybrid CCJT | Improvement factor |
|---|---|---|---|---|
| **Child** | 2 | 67 | 14 | 4.8$\times$ |
| | 4 | 157 | 33 | 4.7$\times$ |
| | 8 | 429 | 58 | 7.4$\times$ |
| **Alarm** | 2 | 166 | 15 | 11.1$\times$ |
| | 4 | 1959 | 40 | 49$\times$ |
| | 8 | 1886 | 79 | 23.9$\times$ |
| **Insurance** | 2 | 143 | 17 | 8.4$\times$ |
| | 4 | 1835 | 43 | 42.7$\times$ |
| | 8 | 473 | 42 | 11.3$\times$ |
| **Andes** | 16 | 23175 | 6080 | 3.8$\times$ |
| **Link** | 64 | 4455 | 459 | 9.7$\times$ |
| **Munin #2** | 128 | 243474 | 735 | 331.3$\times$ |

significantly, reaching as high as 243K transmitted values for Munin. In contrast, `Hybrid CCJT` maintains a low communication overhead, transmitting 735 values ($331\times$ less). As expected, we did not measure any significant difference ($\geq 0.001$) in predictive accuracy.

## 5   Conclusion

Motivated by the need of collaborative and confidential inference in manufacturing settings, we introduce `Hybrid CCJT`, a novel framework enabling collaborative confidential inference on Hybrid Bayesian Networks. By addressing the scalability limitations of existing methods and their inability to model hybrid data, `Hybrid CCJT` facilitates secure collaborative inference while maintaining confidentiality of party models and of private variable posteriors. The proposed framework introduces two pivotal components: a collaborative strongly-rooted JT for constructing a strongly-rooted Junction Tree, and a Confidential Inference Protocol to perform privacy-preserving inference that leverages such a Junction Tree. Our evaluation across nine datasets demonstrates `Hybrid CCJT`'s superior predictive accuracy (by 32% better MSE on average) at reduced communication costs (up to $86\times$ less sent values). In Appendix K we discuss `Hybrid CCJT` limitations, like possible attacks on confidentiality or the ability to handle a broader class of hybrid networks, and possible solutions.

## Acknowledgments

We thank the ASML High-Performance Computing team for enabling some of the analyses conducted in this paper. This research is partly funded by the "Priv-GSyn: Privacy-Preserving Graph Synthesis" grant (200021E_229204) from the Swiss National Science Foundation, the "DEPMAT

---

[2]Timeout per experiment is set to 24 hours, on 512GB RAM.

(Data Enhanced Physical Models to reduce material use)" project (P20-22a) from the "Perspectief" Dutch Research Council (NWO) research programme, and ASM International NV. This work has also been partially supported by the Spoke 1 "FutureHPC & BigData" of the ICSC–Centro Nazionale di Ricerca in High Performance Computing, Big Data and Quantum Computing, and hosting entity, funded by European Union - Next GenerationEU and by the DYMAN project funded by the European Union - European Innovation Council under G.A. n. 101161930.

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

## Nomenclature

$\Delta$      Set of all discrete variables

$\Delta_{\mathcal{L}}^i$      Local discrete clique of party $i$

$\Delta_{\mathcal{I}}$      Interface discrete clique

$\Delta_i$      Set of discrete variables owned by party $i$

$\Gamma$      Set of all continuous variables

$\Gamma_{\mathcal{L}}^i$      Local continuous clique of party $i$

$\Gamma_{\mathcal{I}}$      Interface continuous clique

$\Gamma_i$      Set of continuous variables owned by party $i$

$\mathbb{P}$      Set of the parties

$\mathcal{N}(X; \mu, \sigma)$      Normal distribution over $X$ with mean $\mu$ and standard deviation $\sigma$

$\mathcal{T}$      Set of threshold variables, i.e., $\Delta \cap \mathrm{pa}(\Gamma)$

$\mathcal{U}(S)$      Uniform distribution over set $S$

$\Omega(X)$      Set of states over variable $X$

$\Phi(S)$      Set of factors over variables in $S$

$\phi(X)$      Factor of variable $X$

$\mathrm{CPD}_{i,j}$      Entry $i, j$ of a Conditional Probability Distribution

pa$(X)$  Set of parents of variable $X$

$P[X \mid E = e]$  Probability of variable $X$ conditioned on evidence $E = e$

$P[X \mid E]$  Probability of variable $X$ conditioned on variable $E$

$P[X]$  Probability of variable $X$

# A  Strongly-rooted Junction Tree Proof

Here, we prove the validity of the strongly-rooted Junction Tree $\mathcal{T}$ built following the Collaborative Junction Tree protocol. We first prove that $\mathcal{T}$ is a valid Junction Tree. Then, we proceed to show that any local discrete clique $(\Delta_{\mathcal{L}}^i)$ is a valid strong root for $\mathcal{T}$.

## A.1  Family Preservation Property

**Definition A.1.** A Junction Tree $\mathcal{T}$ over set of factors $\Phi$ is said to be *family preserving* if each factor $\phi \in \Phi$ is associated with one cluster $c_i \in \mathcal{T}$ such that $\text{Scope}[\phi] \subseteq X_i$. With $X_i$ being the set of variables over which $c_i$ is defined.

Since in Hybrid CLG networks, no continuous variable can have a discrete child. It follows that:

**Lemma A.2.** *Scope*$[\Delta] \subseteq \Delta$

Let $\Delta_i$ and $\Gamma_i$ be the set of discrete and continuous variables in party $i$'s input network. It follows that:

**Lemma A.3.** $\forall x \in (\Delta_i \cup \Gamma_i) \setminus \mathcal{I}$ *all parents of $x$ are in* $\Delta_i \cup \Gamma_i$

**Lemma A.4.** $(\Delta \cap \mathcal{I}) \cup (\Gamma \cap \mathcal{I}) = \mathcal{I}$

By definition, $\mathcal{I}$ is the set of all variables that belong to more than one party. From this, Lemma A.5 and Lemma A.6 follow.

**Lemma A.5.** $(\Delta_i \setminus \mathcal{I}) \cap (\Delta_j \setminus \mathcal{I}) = \emptyset \, \forall i, j \in \mathbb{P}$

**Lemma A.6.** $(\Gamma_i \setminus \mathcal{I}) \cap (\Gamma_j \setminus \mathcal{I}) = \emptyset \, \forall i, j \in \mathbb{P}$

Clearly, the set of discrete and continuous variables are disjoint.

**Lemma A.7.** $\Delta \cap \Gamma = \emptyset$

**Proposition A.8.** $\mathcal{T}$ *is family preserving.*

*Proof.* In $\mathcal{T}$ there are four possible types of cliques:

- $\Delta_{\mathcal{L}}^i = (\Delta_i, \Phi(\Delta_i \setminus \mathcal{I}))$

- $\Gamma_{\mathcal{L}}^i = (\Gamma_i \cup \text{Scope}_i(\Gamma_i), \Phi(\Gamma_i \setminus \mathcal{I}))$

- $\Delta_{\mathcal{I}} = ((\text{Scope}[\bigcap_i \Delta_i] \cup (\text{Scope}[\Gamma] \cap \Delta], \Phi(\Delta \cap \mathcal{I}))$

- $\Gamma_{\mathcal{I}} = (\text{Scope}[\bigcap_i \Gamma_i] \cup (\text{Scope}[\Gamma] \cap \Delta, \Phi(\Gamma \cap \mathcal{I}))$

Lemma A.2 $\wedge$ Lemma A.3 $\Rightarrow \text{Scope}[\Delta_i \setminus \mathcal{I}] \subseteq \Delta_i$
$\therefore \Delta_{\mathcal{L}}^i$ is family preserving.

Trivially, $\text{Scope}[\Gamma_i \setminus \mathcal{I}] \subseteq \Gamma_i \cup \text{Scope}[\Gamma_i]$
$\therefore \Gamma_{\mathcal{L}}^i$ is family preserving.

Trivially, $\text{Scope}[\Delta \cap \mathcal{I}] \subseteq (\text{Scope}[\bigcap_i \Delta_i]) \cup (\text{Scope}[\Gamma] \cap \Delta)$
$\therefore \Delta_{\mathcal{I}}$ is family preserving.

Trivially, $\text{Scope}[\Gamma \cap \mathcal{I}] \subseteq \text{Scope}[\bigcap_i \Gamma_i] \cup (\text{Scope}[\Gamma] \cap \Delta$
$\therefore \Gamma_{\mathcal{I}}$ is family preserving.

From Lemma A.4 $\Rightarrow (\bigcup_{i \in \mathbb{P}} \Delta_i \setminus \mathcal{I}) \cup (\bigcup_{i \in \mathbb{P}} \Gamma_i \setminus \mathcal{I}) \cup (\Gamma \cap \mathcal{I}) \cup (\Delta \cap \mathcal{I}) = \Gamma \cup \Delta$
$\therefore$ Each factor is associated with at least one clique.

Lemma A.5 $\Rightarrow (\Delta_i \setminus \mathcal{I}) \cap (\Delta_j \setminus \mathcal{I}) = \emptyset \quad \forall i, j \in \mathbb{P}$
Lemma A.6 $\Rightarrow (\Gamma_i \setminus \mathcal{I}) \cap (\Gamma_j \setminus \mathcal{I}) = \emptyset \quad \forall i, j \in \mathbb{P}$
Trivially, $(\Delta_i \setminus \mathcal{I}) \cap (\Delta \cap \mathcal{I}) = \emptyset \quad \forall i \in \mathbb{P}$
Trivially, $(\Gamma_i \setminus \mathcal{I}) \cap (\Gamma \cap \mathcal{I}) = \emptyset \quad \forall i \in \mathbb{P}$
Lemma A.7 $\Rightarrow (\Gamma \cap \mathcal{I}) \cap (\Delta \cap \mathcal{I}) = \emptyset$
Lemma A.7 $\Rightarrow (\Gamma_i \setminus \mathcal{I}) \cap (\Delta \cap \mathcal{I}) = \emptyset \quad \forall i \in \mathbb{P}$
Lemma A.7 $\Rightarrow (\Delta_i \setminus \mathcal{I}) \cap (\Gamma \cap \mathcal{I}) = \emptyset \quad \forall i \in \mathbb{P}$
Lemma A.7 $\Rightarrow (\Delta_i \setminus \mathcal{I}) \cap (\Gamma_j \setminus \mathcal{I}) = \emptyset \quad \forall i, j \in \mathbb{P}$
$\therefore$ Each factor is associated with at most one clique.

$\therefore \mathcal{T}$ is family preserving. $\hfill\square$

## A.2 Running Intersection Property

We recall all cliques definitions given by construction following the Collaborative Junction Tree protocol as in subsection 3.3. Let $\text{Scope}_i(x)$ be the scope of variable $x$ in party $i$'s input network.

**Lemma A.9.** $\Delta_\mathcal{L}^i = \Delta_i$

**Lemma A.10.** $\Delta_\mathcal{I} = \text{Scope}[\bigcap_i \Delta_i] \cup (\text{Scope}[\Gamma] \cap \Delta)$

**Lemma A.11.** $\Gamma_\mathcal{L}^i = \Gamma_i \cup \text{Scope}_i(\Gamma_i)$

**Lemma A.12.** $\Gamma_\mathcal{I} = \text{Scope}[\bigcap_i \Gamma_i] \cup (\text{Scope}[\Gamma] \cap \Delta)$

We recall the definition of the running intersection property:

**Definition A.13.** A Junction Tree $\mathcal{T}$ satisfies the running intersection property if, for all pairs of cliques $(\mathcal{C}_1, \mathcal{C}_2)$ in $\mathcal{T}$, $\mathcal{C}_1 \cap \mathcal{C}_2$ is also in every clique in the path in $\mathcal{T}$ between $\mathcal{C}_1$ and $\mathcal{C}_2$.

From Definition A.13, it trivially follows that:

**Lemma A.14.** *Every pair of neighboring cliques satisfy the running intersection property.*

**Proposition A.15.** $\mathcal{T}$ *satisfies the running intersection property.*

*Proof.* By Lemma A.14, all pairs of neighboring cliques satisfy the running intersection property. In $\mathcal{T}$, there are five possible pairs of non-neighboring cliques:

- $\mathcal{P}_1 = (\Delta_\mathcal{L}^i, \Delta_\mathcal{L}^j)$, with $\Delta_\mathcal{I}$ the only clique in the path.

- $\mathcal{P}_2 = (\Gamma_\mathcal{L}^i, \Gamma_\mathcal{L}^j)$, with $\Gamma_\mathcal{I}$ the only clique in the path.

- $\mathcal{P}_3 = (\Delta_\mathcal{L}^i, \Gamma_\mathcal{I})$, with $\Delta_\mathcal{I}$ the only clique in the path.

- $\mathcal{P}_4 = (\Gamma_\mathcal{L}^i, \Delta_\mathcal{I})$, with $\Gamma_\mathcal{I}$ the only clique in the path.

- $\mathcal{P}_5 = (\Delta_\mathcal{L}^i, \Gamma_\mathcal{L}^j)$, with $\Delta_\mathcal{I}$ and $\Gamma_\mathcal{I}$ in the path.

Lemma A.10 $\Rightarrow \Delta_\mathcal{L}^i \cap \Delta_\mathcal{L}^j \subseteq \Delta_\mathcal{I}$
$\therefore \mathcal{P}_1$ satisfies the running intersection property.

Lemma A.12 $\Rightarrow \Gamma_\mathcal{L}^i \cap \Gamma_\mathcal{L}^j \subseteq \Gamma_\mathcal{I}$
$\therefore \mathcal{P}_2$ satisfies the running intersection property.

Lemma A.9 $\wedge$ Lemma A.12 $\Rightarrow \Delta_\mathcal{L}^i \cap \Gamma_\mathcal{I} \subseteq \text{Scope}[\Gamma] \cap \Delta$
Lemma A.10 $\Rightarrow \text{Scope}[\Gamma] \cap \Delta \subseteq \Delta_\mathcal{I}$
$\therefore \mathcal{P}_3$ satisfies the running intersection property.

Lemma A.11 $\wedge$ Lemma A.10 $\Rightarrow \Gamma_\mathcal{L}^i \cap \Delta_\mathcal{I} \subseteq \text{Scope}[\Gamma] \cap \Delta$
Lemma A.10 $\Rightarrow \text{Scope}[\Gamma] \cap \Delta \subseteq \Gamma_\mathcal{I}$
$\therefore \mathcal{P}_4$ satisfies the running intersection property.

Lemma A.9 $\wedge$ Lemma A.11 $\Rightarrow \Delta_\mathcal{L}^i \cap \Gamma_\mathcal{L}^i \subseteq \text{Scope}[\Gamma] \cap \Delta$
Lemma A.10 $\Rightarrow \Delta_\mathcal{I} \subseteq \text{Scope}[\Gamma] \cap \Delta$
Lemma A.12 $\Rightarrow \Gamma_\mathcal{I} \subseteq \text{Scope}[\Gamma] \cap \Delta$
$\therefore \mathcal{P}_5$ satisfies the running intersection property.

$\therefore \mathcal{T}$ satisfies the running intersection property. □

## A.3  Strong Root Property

Following Definition 3.1, we prove that $\Delta_{\mathcal{L}}^i$ is a strong root of $\mathcal{T}$.

From Lemma A.11 and Lemma A.12, it follows that:

**Lemma A.16.** $\Gamma_{\mathcal{L}}^i \cap \Delta \subseteq \Gamma_{\mathcal{I}} \cap \Delta$

From Lemma A.10 and Lemma A.2, it follows:

**Lemma A.17.** $\Delta_{\mathcal{I}} \subseteq \Delta$

**Proposition A.18.** $\Delta_{\mathcal{L}}^i$ *is a strong root of* $\mathcal{T}$.

*Proof.* In $\mathcal{T}$ there are three possible pairs of neighboring cliques:

- $\mathcal{S}_1$: $(\Gamma_{\mathcal{L}}^i, \Gamma_{\mathcal{I}})$, with $\Gamma_{\mathcal{I}}$ being closer to the root.

- $\mathcal{S}_2$: $(\Gamma_{\mathcal{I}}, \Delta_{\mathcal{I}})$, with $\Delta_{\mathcal{I}}$ being closer to the root.

- $\mathcal{S}_3$: $(\Delta_{\mathcal{L}}^i, \Delta_{\mathcal{I}})$, where either can be closer to the root depending on which party requests the query.

Lemma A.16 $\Rightarrow (\Gamma_{\mathcal{L}}^i \setminus \Gamma_{\mathcal{I}}) \cap \Delta = \emptyset \Rightarrow \Gamma_{\mathcal{I}} \setminus \Gamma_{\mathcal{L}}^i \subseteq \Gamma$
$\therefore \mathcal{S}_1 \models$ Equation 3.

Lemma A.17 $\Rightarrow \Delta_{\mathcal{I}} \cap \Gamma_{\mathcal{I}} \subseteq \Delta$
$\therefore \mathcal{S}_2 \models$ Equation 3

Lemma A.17 $\Rightarrow \Delta_{\mathcal{L}}^i \cap \Delta_{\mathcal{I}} = \Delta_{\mathcal{I}} \cap \Delta_{\mathcal{L}}^i \subseteq \Delta$
$\therefore \mathcal{S}_3 \models$ Equation 3

$\therefore \Delta_{\mathcal{L}}^i$ is a strong root of $\mathcal{T}$. □

# B  Confidentiality of Hybrid CCJT

Here, we present the achievements in confidentiality preservation of `Hybrid CCJT`. Before proceeding to the discussion, we introduce some auxiliary concepts.

**Confidentiality in Bayesian Networks** When dealing with confidentiality in Bayesian Networks it is worth noticing that some information leakage is typically inevitable. In fact, simply knowing the result of an inference query could be used to reverse-compute further information. For instance, [30] studied privacy-preserving applications of algorithms such as Belief Propagation and Sampling. In Kearns' application, messages in the Belief Propagation process must be protected. However, leaf nodes can always reverse compute incoming messages from their neighbors from solely the inference outcome [30]. To formalize this notion, [30] give the following definition of a privacy-preserving protocol:

**Definition B.1.** Let $\Pi$ be any protocol for the $k$ parties to jointly compute the value $y = f(x_1, ..., x_k)$ from their $n$-bit private inputs. We say that $\Pi$ is privacy-preserving if for every $1 \leq i \leq k$, anything that party $i$ can compute in polynomial time in $n$ following the execution of $\Pi$, they could also compute in polynomial time given only their private input $x_i$ and the value $y$.

In this section, we show that `Hybrid CCJT` abides Definition B.1. That is, no information about *structure*, *parameters*, and *posteriors* of other parties' BNs are revealed after performing a query with `Hybrid CCJT`.

## B.1  Structure and Parameters

**Junction Tree Construction** `Hybrid CCJT` defines a collaborative junction tree without requiring any information disclosure between the parties. In fact, every party knows which of their variables to

allocate in the local and interface cliques. This is sufficient to carry out inference as in our protocol, but no party knows anything about the content nor the structure of other parties' local cliques.

**Discrete Factor Alignment** Parties' factors are updated for discrete interface factors only. As anticipated in subsection 3.3, this is done via securely computing the column normalization factor $\alpha$. Let $\text{CPD}_{i,j}$ denote party $i$ CPD column $j$, then $\alpha = || \odot_i \text{CPD}_{i,j}||_1$ is computed via homomorphic encryption. Our implementation relies on the CKKS [20] scheme for floating-point addition and multiplication. This procedure allows to apply the geometric mean correctly without disclosing the parties' CPD entries. As a matter of fact, after computing the (scalar) normalization factor $\alpha$, it is not possible to decompose it into the $|\text{CPD}_{i,j}| \propto 2^N$ values used to compute it in polynomial time, abiding to Definition B.1.

## B.2 Posterior of Private Variables

`Hybrid CCJT` executes two subroutines to perform collaborative inference: one on the *discrete domain*, one on the *continuous domain*. We show that no party gains any information about the posterior of other parties' private variables (i.e., that are not part of the interface).

**Posterior of Discrete Variables** We ensure that only the querying party handles the normalization process of interface factors. As shown in Equation 9, this allows each party to marginalize all variables that are not shared with the querying party prior to message passing. As such, the merging procedure will only reveal the posterior of the interface variables, which are owned by the querying party.

Let $X$ be a discrete variable owned by two parties $A$ and $B$, with $A$ being the querying party.
Let $\text{pa}_i(X)$ be the set of parent variables of $X$ owned by party $i$.
Let $P_A[X]$, $P_A'[X]$ be the probability distribution according to $A$ before and after the message passing procedure respectively.

We show that after performing factor alignment (Proposition B.2) and message-passing (Proposition B.5), the querying party does not gain any additional information about the true probability distribution of variables in $\text{pa}_B(X)$.

**Proposition B.2.** $P_A[pa_B(X)] = \mathcal{U}\{\Omega(pa_B(X))\}$

*Proof.* By construction, $P_A$ is obtained by duplicating each entry to allocate space for variables in $\text{pa}_B(X)$. It follows that:

$$\sum_{var \notin \text{pa}_B(X)} P_A[\text{pa}_B(X) = i] = \frac{1}{|\Omega(\text{pa}_B(X))|} \quad \forall i \in \Omega(\text{pa}_B(X)) \tag{11}$$

Thus,

$$P_A[\text{pa}_B(X)] = \mathcal{U}\{\Omega(\text{pa}_B(X))\} \tag{12}$$

$\square$

Let $S = \text{pa}_A(X) \cup \text{pa}_B(X) \cup \{X\}$.
Let $S' = S \setminus \text{pa}_B(X)$.
The following is a corollary of Proposition B.2:
**Corollary B.3.** $\sum_{j \in S'} P_A[S = j, i] = \frac{1}{|\Omega(pa_B(X))|} \quad \forall i \in \Omega(pa_B(X))$

From Equation 2, it follows:
**Lemma B.4.** $P_A'[S] = P_A[S] \cdot \sum_{var \in pa_B(X)} P_B[S]$

Let $P_B' = \sum_{var \in \text{pa}_X(B)} P_B$
**Proposition B.5.** $P_A[pa_B(X)] = \mathcal{U}\{\Omega(pa_B(X))\} \Rightarrow P_A'[pa_B(X)] = \mathcal{U}\{\Omega(pa_B(X))\}$

*Proof.*

$$P_A'[\text{pa}_B(X) = i] = \frac{\sum_{j \in \Omega(S):\text{pa}_B(X)=i} P_A'[S = j]}{\sum_{k \in \Omega(S)} P_A'[S = k]} \quad \forall i \in \Omega(\text{pa}_B(X)) \tag{13}$$

$$= \frac{\sum_{j \in \Omega(S')} \left\{ \begin{array}{c} P_A[S = j, i] \cdot \\ \sum_{l \in \Omega(\mathrm{pa}_B(X))} P_B[S = j, l] \end{array} \right\}}{\sum_{k \in \Omega(S')} \left\{ \begin{array}{c} \sum_{m \in \Omega(\mathrm{pa}_B(X))} P_A[S = k, m] \cdot \\ \sum_{n \in \Omega(\mathrm{pa}_B(X))} P_B[S = k, n] \end{array} \right\}} \qquad \forall i \in \Omega(\mathrm{pa}_B(X)) \qquad (14)$$

$$= \frac{\sum_{j \in \Omega(S')} \left\{ \begin{array}{c} \frac{1}{|\Omega(\mathrm{pa}_B(X))|} P_A[S' = j] \cdot \\ P'_B[S' = j] \end{array} \right\}}{\sum_{k \in \Omega(S')} \left\{ \begin{array}{c} P_A[S' = k] \cdot \\ P'_B[S' = k] \end{array} \right\}} \qquad \forall i \in \Omega(\mathrm{pa}_B(X)) \qquad (15)$$

$$= \frac{1}{|\Omega(\mathrm{pa}_B(X))|} \cdot \frac{\sum_{j \in \Omega(S')} \left\{ \begin{array}{c} P_A[S' = j] \cdot \\ P'_B[S' = j] \end{array} \right\}}{\sum_{k \in \Omega(S')} \left\{ \begin{array}{c} P_A[S' = k] \cdot \\ P'_B[S' = k] \end{array} \right\}} \qquad \forall i \in \Omega(\mathrm{pa}_B(X)) \qquad (16)$$

$$= \frac{1}{|\Omega(\mathrm{pa}_B(X))|} \qquad \forall i \in \Omega(\mathrm{pa}_B(X)) \qquad (17)$$

Thus,

$$P'_A[\mathrm{pa}_B(X)] = \mathcal{U}\{\Omega(\mathrm{pa}_B(X))\} \qquad (18)$$

$\square$

**Generalization to $N$ parties** Proposition B.5 can be generalized to $N$ parties by substituting $\sum_{l \in \Omega(\mathrm{pa}_B(X))} P_B[S = j, l]$ with $\prod_{i \in \mathbb{P} \setminus \{A\}} \sum_{l \in \Omega(\mathrm{pa}_i(X))} P_i[S = j, l]$ and $P'_B$ with $\prod_{i \in \mathbb{P} \setminus \{A\}} P'_i$.

**Posterior of Continuous Variables** As anticipated in subsection 3.3, continuous variables mean and variances are computed via multi-party secret sharing updating the parameters of continuous interface variables. This process inherently protects private variables' information as they are not involved in this update operation.

## C  Canonical Representation of Conditional Linear Gaussian CPDs

A common representation used for CLG BNs is the *canonical form* [7], which represents CLG factors as a log-quadratic form $\exp(Q(X))$ where $Q$ is some quadratic function.

**Definition C.1.** A canonical form $\mathcal{C}(X; K, h, g)$ is defined as

$$\mathcal{C}(X; K, h, g) = \exp\left(-\frac{1}{2} X^T K X + h^T X + g\right) \qquad (19)$$

We can represent every Gaussian in the canonical form [7]:

$$\mathcal{N}(\mu; \Sigma) = \mathcal{C}(K, h, g) \qquad (20)$$

where

$$K = \Sigma^{-1}, \quad h = \Sigma^{-1}\mu, \quad g = -\frac{1}{2}\mu^T \Sigma^{-1}\mu - \log\left((2\pi)^{n/2}|\Sigma|^{1/2}\right) \qquad (21)$$

### C.1  Operations on Canonical Form

**Factor Product** The product of two canonical forms over the same scope can be easily computed as:

$$\mathcal{C}(K_1, h_1, g_1) \cdot \mathcal{C}(K_2, h_2, g_2) = \mathcal{C}(K_1 + K_2, h_1 + h_2, g_1 + g_2) \qquad (22)$$

Whenever two factors are not defined over the same scope, they can be expanded by simply padding the $K, h$ parameters with zeros.

**Factor Marginalization** We can marginalize a canonical form onto a subset of its variables. Let $\mathcal{C}(X, Y; K, h, g)$ be some canonical form over $\{X, Y\}$ where:

$$K = \begin{bmatrix} K_{XX} & K_{XY} \\ K_{YX} & K_{YY} \end{bmatrix}, \quad h = \begin{bmatrix} h_X \\ h_Y \end{bmatrix} \qquad (23)$$

The marginalized factor $\mathcal{C}(X; K', h', g') = \int \mathcal{C}(X, Y; K, h, g)\, dY$ is computed as:

$$K' = K_{XX} - K_{XY}K_{YY}^{-1}K_{YX}, \quad h' = h_X - K_{XY}K_{YY}^{-1}h_Y,$$
$$g' = g + \frac{1}{2}\left(\log|2\pi K_{YY}^{-1}| + h_Y^T K_{YY}^{-1}h_Y\right) \tag{24}$$

**Factor Reduction** Finally, it is possible to reduce a canonical form based on a certain evidence. Let $\mathcal{C}(X, Y; K, h, g)$ be some canonical form over $\{X, Y\}$ represented as in Equation 23. Then, setting $Y = y$ results in the canonical form $\mathcal{C}(X; K', h', g')$, where:

$$K' = K_{XX}, \quad h' = h_X - K_{XY}y, \quad g' = g + h_Y^T y - \frac{1}{2}y^T K_{YY}y \tag{25}$$

**Computational Costs** Importantly, all factor operations can be done in polynomial time in the scope of the factor. In particular, the product or division of factors requires quadratic time. Factor marginalization, which requires matrix inversion, can be done naively in $\mathcal{O}(N^3)$, and more efficiently using advanced methods.

### C.2  Canonical Form in Hybrid CLG BNs

In hybrid BNs, using both discrete ($\Delta$) and continuous ($\Gamma$) variables, factors with both types of variables can be represented using a *canonical table* [7].

**Definition C.2.** A canonical table $\phi$ over $D, X$ with $D \subseteq \Delta$ and $X \subseteq \Gamma$ is a table with an entry for each $d \in \Omega(D)$ where each entry contains a canonical form $\phi(d) = \mathcal{C}(X; K_d, h_d, g_d)$.

Note that the canonical table representation subsumes both the canonical form and the table factors used in the context of discrete networks. For the former, $D = \emptyset$, we have only a single canonical form over $X$. For the latter, $X = \emptyset$, the parameters $K_d, h_d$ are vacuous, and we remain with a canonical form $\phi(d) = \exp(g_d)$ for each entry.

## D  Strong Elimination Order Example

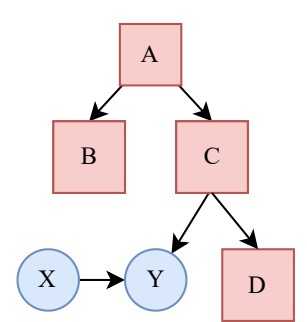

(a) An example Bayesian Network

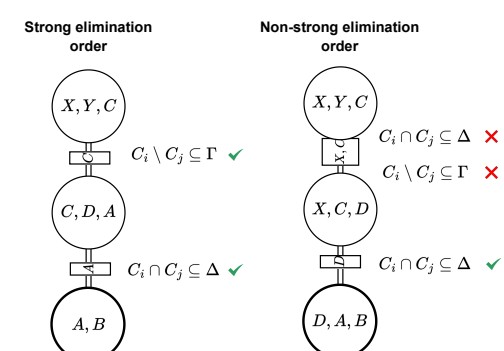

(b) Two possible elimination orders over the network. Root cliques are highlighted with thicker borders.

Figure 4: (a) shows an example Bayesian Network. (b) shows two possible elimination orders over it. Only the left one is strongly-rooted, allowing accurate inference in hybrid domains.

## E  Collaborative Inference for Discrete Query Messages

As introduced in Section 3, when performing collaborative confidential inference (Algorithm 3), we aim at finding $\mu_{\Delta_\mathcal{I} \to \Delta_\mathcal{L}^Q}$. Following the sum-product message-passing formulation (Equation 2) [10]:

$$\mu_{\Delta_\mathcal{I} \to \Delta_\mathcal{L}^Q} = \sum_{x \notin \Delta_Q} \phi(\Delta_\mathcal{I}) \prod_{i \in P \setminus \{Q\}} \mu_{\Delta_\mathcal{L}^i \to \Delta_\mathcal{I}} \tag{26}$$

We follow the formulation based on the geometric mean for $\phi(\Delta_\mathcal{I})$ (Equation 5) and rewrite the equation:

$$\mu_{\Delta_\mathcal{I} \to \Delta_\mathcal{L}^Q} = \sum_{x \notin \Delta_Q} \alpha \prod_{i \in P} \phi(\Delta_\mathcal{I}^i)^{w_i} \prod_{i \in P \setminus \{Q\}} \mu_{\Delta_\mathcal{L}^i \to \Delta_\mathcal{I}} \tag{27}$$

$$= \sum_{x \notin \Delta_Q} \alpha \phi(\Delta_{\mathcal{I}}^Q)^{w_Q} \prod_{i \in P \setminus \{Q\}} \phi(\Delta_{\mathcal{I}}^i)^{w_i} \mu_{\Delta_{\mathcal{L}}^i \to \Delta_{\mathcal{I}}} \tag{28}$$

Since the marginalization operator does not require to eliminate any variable in $\Delta_Q$, we can rewrite the equation as:

$$\mu_{\Delta_{\mathcal{I}} \to \Delta_{\mathcal{L}}^Q} = \alpha \phi(\Delta_{\mathcal{I}}^Q)^{w_q} \sum_{x \notin \Delta_Q} \prod_{i \in P \setminus \{Q\}} \phi(\Delta_{\mathcal{I}}^i)^{w_i} \mu_{\Delta_{\mathcal{L}}^i \to \Delta_{\mathcal{I}}} \tag{29}$$

Furthermore, when multiplying CPD tables, variables that are not common in both tables can be marginalized prior to the product without affecting the result.

$$\mu_{\Delta_{\mathcal{I}} \to \Delta_{\mathcal{L}}^Q} = \alpha \phi(\Delta_{\mathcal{I}}^Q)^{w_Q} \prod_{i \in P \setminus \{Q\}} \sum_{x \notin \Delta_Q} \phi(\Delta_{\mathcal{I}}^i)^{w_i} \mu_{\Delta_{\mathcal{L}}^i \to \Delta_{\mathcal{I}}} \tag{30}$$

## F   Results on Continuous Data

Table 4: Result on continuous data: mean squared errors.

| Dataset | | Ecoli70 | Magic-Niab |
|---|---|---|---|
| **Hybrid CCJT** | | **0.070** | **0.287** |
| | 3 States | 1.411 | 0.521 |
| **Discretized BN** | 5 States | 0.367 | 0.533 |
| | 10 States | 0.368 | N/A |

Here, we extend our experiments on purely continuous datasets, i.e., Ecoli70 and Magic-Niab and summarize the results in Table 4. Since these datasets do not contain any categorical variable, we only run collaborative continuous inference as shown in Algorithm 2. Our baseline is the exact inference on discretized datasets with different levels of coarseness. Due to poor scalability of CCBNet and $\Delta$-CCJT when dealing with discretized datasets, we use a centralized discrete network instead with no communication cost. This provides an upper bound on the predictive performance of a discretization-based approach. Under both experiments, we ran 10000 queries, with 4 parties and 10% overlap.

Across all experiments, `Hybrid CCJT` achieves the highest predictive accuracy. Under the Ecoli70 dataset, `Hybrid CCJT` attains an MSE of 0.07, which is five times lower than the optimal discrete counterpart and 20 times better than the non-optimal one. Under the Magic-Niab dataset, our model achieves an MSE of 0.287, outperforming the discretized counterpart, which has an MSE of 0.521. Furthermore, the discretized model failed to run inference in a reasonable amount of time with 10 states on Magic-Niab.

## G   Datasets Details

Table 5: Overview of used datasets.

| Type | Dataset | #Discrete nodes | #Continuous nodes | #Arcs | #Params | Source |
|---|---|---|---|---|---|---|
| Hybrid CLG | Healthcare | 3 | 4 | 9 | 42 | [23] |
| | Sangiovese | 1 | 14 | 55 | 259 | [31] |
| | Mehra | 8 | 16 | 71 | 324423 | [32] |
| Discrete | Child | 20 | - | 25 | 230 | [33] |
| | Alarm | 37 | - | 46 | 509 | [34] |
| | Insurance | 27 | - | 52 | 1008 | [35] |
| | Andes | 223 | - | 338 | 1157 | [36] |
| | Link | 724 | - | 1125 | 14211 | [37] |
| | Munin #2 | 1003 | - | 1244 | 69431 | [38] |
| Continuous | Ecoli70 | - | 46 | 70 | 162 | [39] |
| | Magic-Niab | - | 44 | 66 | 154 | [40] |

In Table 5, we provide the characteristics about the datasets used in our experiments. We used three Hybrid CLG datasets (Healthcare, Sangiovese, Mehra), six discrete datasets (Child, Alarm, Insurance, Andes, Link, Munin #2), and two CLG ones (Ecoli70, Magic-Niab).

## H Hybrid CCJT **Diagram**

To graphically showcase the `Hybrid CCJT` inference pipeline, Figure 5 and 6 provide a detailed flowchart of collaborative inference with two parties.

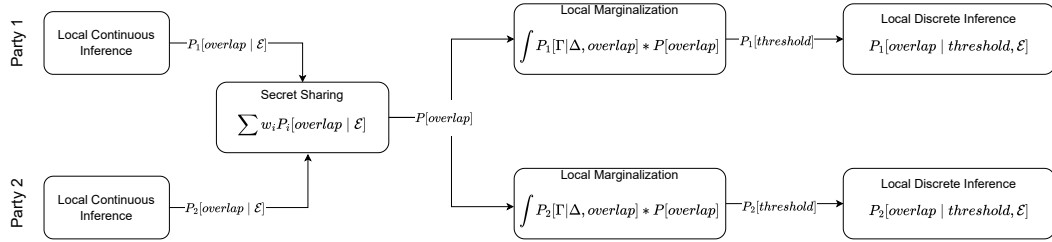

Figure 5: Flowchart of `Hybrid CCJT` collaborative continuous inference procedure with two parties.

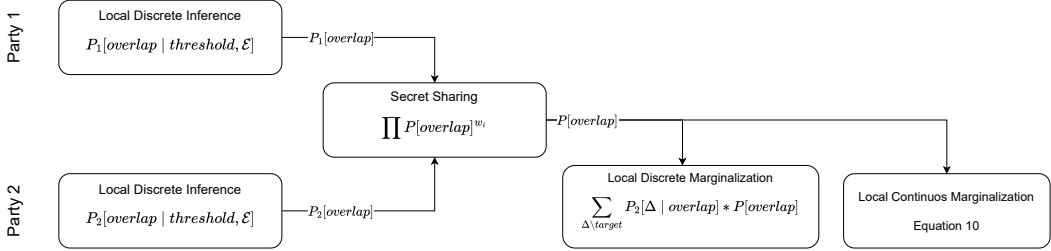

Figure 6: Flowchart of `Hybrid CCJT` collaborative discrete inference procedure with two parties. We will prepare a more detailed explanation of the inference protocol in the methodology section and a more detailed pseudocode and a diagram to be included in the appendix. The inference is divided in three steps (of which two are collaborative) and follows the logic of the Lauritzen algorithm [11].

**Collaborative Continuous Inference** The first step requires marginalizing continuous variables and finding the **strong marginals** over the threshold variables. Recall that threshold variables are the discrete variables with at least one continuous child. In short, we aim at computing $P[T|\Gamma = e]$, where $T$ is the set of threshold variables and $\Gamma = e$ is some continuous evidence. To do so, parties collaboratively compute the updated parameters of interface continuous variables given the set of continuous evidences as in

$$\mu(X) = \sum_{i \in \mathbb{P}} w_i \mu_i(X), \quad \sigma(X) = \sum_{i \in \mathbb{P}} [w_i \sigma_i(X)] + \sum_{i \in \mathbb{P}} [w_i (\mu - \mu_i(X))^2]$$

where the parameters $\mu_i$ and $\sigma_i$ are obtained locally by computing $P[I_\Gamma | \Gamma = e]$. Once these new parameters are obtained, they can help compute the marginals over the threshold variables by just marginalizing all the continuous variables.

**Collaborative Discrete Inference** Once parties find the marginals of threshold variables, the querying party needs to find the strong marginals over the remaining discrete variables. For this, parties compute $P[I_\Delta | \Delta = e]$, the posterior of discrete interface variables given their discrete evidence. The querying party receives these posteriors and merge them via secret sharing to find the geometric mean:

$$P[I_\Delta] = \prod_{i \in \mathbb{P}} P_i[I_\Delta]^{w_i}$$

The querying party can then use this factor to update its local probability distribution via sum-product message passing [10] (Equation 2).

**Local Query** At this point in the protocol, the querying party can find the posterior of any variable it owns. For **discrete variables**, the posterior is found by marginalizing non-target variables, as in a normal junction tree. For **continuous variables**, the posterior is found by marginalizing all non-threshold variables and performing weak marginalization:

$$P[X] = \mathcal{N}\Big(X; \sum_{s \in \Omega(\Delta_Q)} P[\mathcal{T}_Q = s]\mu_s(X), \sum_{s \in \Omega(\Delta_Q)} P[\mathcal{T}_Q = s]\{\sigma_s(X) + (\mu - \mu_s(X))^2\}\Big)$$

# I  Computational Costs of Cryptographic Tools

The computation complexity of homomorphic encryption in solving an overlap is $O(L \cdot N \log(N))$, bounded by $L$ chained multiplications of a size $N$ ciphertext. $L$ is the maximum number of parties in an overlap. $N$ is a power of two (in the range $2^{10}$–$2^{15}$), derived from the desired precision, the maximum output scale, and $L$. Successive multiplications increase complexity linearly [20]. The cost of a single multiplication is linearithmic [41]. In our implementation, solving a private set intersection between two parties is $O(m \log(m))$ time, where $m$ is the maximum number of variables among the two parties.

The protocol relies on private set intersection, secret sharing (additive/product-based), and homomorphic encryption (HE) at different points in its lifecycle. Among the enumerated cryptographic tools, only HE involves a significant computational overhead, while the primary bottleneck for the others is network latency. HE is used exclusively during discrete factor alignment. The results below split the HE computation time across:

1. *Context Generation*: Generation of HE keys and parameters.

2. *Encrypted Column Summation*: The actual HE-encrypted computation; used to compute column normalization factors (i.e., $\alpha$ in Equation 4).

3. *CPD Combination*: Alignment of party CPDs based on the HE-computation outputs.

We report the **per-factor mean** and **standard deviation** of each HE substep and the total time required for all factors.

Table 6: Results on discrete datasets (10% overlap)

| Dataset | #Parties | Context Gen. (ms) | Col. Sums (ms) | CPD comb. (ms) | Total (ms) |
|---|---|---|---|---|---|
| Asia | 2 | 153± 0 | 18.7± 0 | 0.27± 0 | 176 |
| | 4 | 644± 0 | 802± 0 | 0.53± 0 | 1449 |
| | 8 | 2155± 0 | 767± 0 | 0.746± 0 | 2926 |
| Alarm | 2 | 136± 5.7 | 51401± 82611 | 9.57± 14.6 | 206191 |
| | 4 | 672± 50 | 3639± 4780 | 0.749± 0.4 | 17250 |
| | 8 | 1900± 150 | 96745± 134754 | 4.19± 4.69 | 394601 |
| Child | 2 | 143± 5 | 144± 84 | 0.33± 0.034 | 577 |
| | 4 | 703± 26 | 17151± 9176 | 1.78± 0.6 | 35713 |
| | 8 | 1984± 57 | 83010± 79570 | 3.9± 2.9 | 169999 |
| Link | 64 | 133± 7 | 324.9± 381 | 0.4± 0.1 | 431291 |

As expected, substep 2, i.e., encrypted column summation, is the most time-consuming one, taking up to 96 seconds in the worst-case scenario, but remains under one second in 6 out of 10 experiment scenarios. Its time requirements increase overall with the number of parties sharing factors. In contrast, substeps 1 and 3, i.e., context generation and CPD combination, run in under one second and one millisecond, respectively. Moreover, we note that the HE procedure represents a one-off setup cost, after which parties may perform unlimited queries until one of them updates their local network.

When dealing with hybrid domains, we notice that encryption takes significantly more time on the discretized variants. While `Hybrid CCJT` maintains the entire encryption process under one second for both Sangiovese and Healthcare datasets, $\Delta$-CCJT's encryption cost grows as the discretization coarseness is reduced (i.e., more states). These results also reaffirm the gains in memory and communication costs of natively handling CLG variables compared to discretizing them.

# J  Communication Costs Analysis

We repeated the experiments from Table 1 with new variable splits to derive further insights in the communication costs of `Hybrid CCJT`. Table 8 isolates the communication costs of Discrete and Continuous collaborative inference. This allows us to better analyze the communication costs

Table 7: Results on hybrid datasets (30% overlap)

| Dataset | #Parties | Context Gen. (ms) | Col. Sums (ms) | CPD comb. (ms) | Total (ms) |
|---|---|---|---|---|---|
| Healthcare (Hybrid) | 2 | $127\pm 0$ | $26\pm 0$ | $0.23\pm 0$ | 154 |
| Healthcare (Discrete, 3 states) | 2 | $142\pm 8.24$ | $82.4\pm 67$ | $0.3\pm 0.05$ | 675 |
| Healthcare (Discrete, 5 states) | 2 | $71.35\pm 13$ | $71.4\pm 62.4$ | $0.271\pm 0.03$ | 641 |
| Healthcare (Discrete, 10 states) | 2 | $139.7\pm 10.2$ | $165.5\pm 152.6$ | $0.316\pm 0.04$ | 918 |
| Healthcare (Hybrid) | 4 | $646.9\pm 6.04$ | $244\pm 90.5$ | $0.43\pm 0.02$ | 1785 |
| Healthcare (Discrete, 3 states) | 4 | $678.46\pm 49.3$ | $300.2\pm 139.2$ | $0.48\pm 0.06$ | 2940 |
| Healthcare (Discrete, 5 states) | 4 | $682.9\pm 46.7$ | $3007.8\pm 3660$ | $0.662\pm 0.295$ | 11076 |
| Healthcare (Discrete, 10 states) | 4 | $677\pm 51$ | $5956\pm 7221.4$ | $0.86\pm 0.48$ | 19905 |
| Sangiovese (Hybrid) | 2 | $127\pm 0$ | $12.1\pm 0$ | $0.25\pm 0$ | 141 |
| Sangiovese (Discrete, 3 states) | 2 | $141\pm 9.7$ | $827\pm 1080$ | $0.46\pm 0.24$ | 2906 |
| Sangiovese (Discrete, 5 states) | 2 | $139.8\pm 9.2$ | $710.3\pm 905$ | $0.403\pm 0.18$ | 2553 |
| Sangiovese (Discrete, 10 states) | 2 | $141\pm 11.9$ | $621\pm 489$ | $0.424\pm 0.09$ | 2289 |
| Sangiovese (Hybrid) | 4 | $127\pm 0$ | $12.1\pm 0$ | $0.25\pm 0$ | 140 |
| Sangiovese (Discrete, 3 states) | 4 | $139\pm 9.7$ | $813.7\pm 1063$ | $0.43\pm 0.24$ | 2861 |
| Sangiovese (Discrete, 5 states) | 4 | $659.3\pm 49.6$ | $188826.7\pm 239278.3$ | $13.2\pm 16.4$ | 758015 |
| Sangiovese (Discrete, 10 states) | 4 | $838\pm 51$ | $1082339\pm 1591371$ | $51\pm 72.8$ | 4332927 |

gains obtained by natively handling continuous data. Furthermore, we report standard deviation measurement of communication costs.

Table 8: Communication costs analysis for discrete and continuous variables

| Dataset | | Healthcare | | | | Sangiovese | | | |
|---|---|---|---|---|---|---|---|---|---|
| #Parties | | 2 | 2 | 4 | 4 | 2 | 2 | 4 | 4 |
| Overlap | | 10% | 30% | 10% | 30% | 10% | 30% | 10% | 30% |
| HybridCCJT | Discrete | $11\pm 4.2$ | $5\pm 4.4$ | $20\pm 11.6$ | $48\pm 43.9$ | $32\pm 0$ | $32\pm 0$ | $96\pm 0$ | $96\pm 0$ |
| | CLG | $0\pm 0$ | $21\pm 4.8$ | $0\pm 0$ | $192\pm 62.1$ | $29\pm 7.5$ | $59\pm 10.5$ | $182\pm 44.9$ | $541\pm 73.6$ |
| $\Delta$-CCJT | 3 States | $11\pm 3.9$ | $21\pm 11.2$ | $14\pm 5.6$ | $70\pm 24.2$ | $21\pm 1.0$ | $177\pm 37.9$ | $243\pm 45.0$ | $3313\pm 1251.6$ |
| | 5 States | $10\pm 3.8$ | $20\pm 6.7$ | $10\pm 1.6$ | $121\pm 78.8$ | $38\pm 1.1$ | $225\pm 36.0$ | $279\pm 46.7$ | $35591\pm 19277.8$ |
| | 10 States | $8\pm 3.1$ | $39\pm 15$ | $10\pm 1.8$ | $229\pm 148.7$ | $27\pm 2.2$ | $334\pm 146.0$ | $539\pm 100.2$ | $5131\pm 1842.4$ |
| CCBNet | 3 States | $58\pm 27.3$ | $61\pm 15.6$ | $20\pm 5.0$ | $120\pm 30.6$ | $240\pm 77.1$ | $1021\pm 298.3$ | $25609\pm 11412.6$ | $28505\pm 11896.8$ |
| | 5 States | $14.7\pm 3.7$ | $72\pm 29.8$ | $10\pm 1.6$ | $888\pm 428.7$ | $55\pm 7.0$ | $1557\pm 506.7$ | $5210\pm 1971.2$ | $271053\pm 125799.6$ |
| | 10 States | $11\pm 3.2$ | $290\pm 139$ | $10\pm 1.8$ | $2068\pm 1043$ | $169\pm 40.8$ | $990\pm 247.0$ | $4766\pm 1438.7$ | $577838\pm 295695.1$ |

# K  Limitations and Future Work

**Attacks on Hybrid CCJT** When performing inference with `Hybrid CCJT` with $N$ parties, secret shared messages can be reverse computed if $N - 1$ parties collude. This means that this kind of information leakage is inevitable when only two parties participate in the inference process. For instance, let us look at secret sharing for continuous variables. The merged posterior of common continuous variables is a weighted average of parties' locally computed posteriors. Then, one party can reverse-compute its peer posterior using the secret sharing outcome and its own locally computed posterior. Similarly, the same goal can be achieved when secret sharing posteriors of common discrete variables, dividing the outcome by the CPD table of the locally computed posterior.

These two attacks abide our security definition, as structure and parameters of the local Bayesian Networks remain protected. Furthermore, the attack previously mentioned falls under Definition B.1 of a privacy-preserving protocol, as the secret local posterior is found using the outcome of the protocol and prior knowledge. In order to avoid this information leakage, it is possible to limit the outcome detail of a query. *Maximum A Posteriori* (MAP) queries [7] allow to find the most-likely state combination of a set of variables given a set of evidence instead of revealing their posterior distribution. Conveniently, MAP queries can be performed via Belief Propagation on a Junction Tree using the Max-Product message-passing procedure [7]. Future research will focus on applying MAP queries to collaborative Junction Trees.

**Finding Common Nodes** `Hybrid CCJT` assumes that the same variable is given the same name across multiple parties. This allows us to use Private Set Intersection to find common nodes among parties' networks. On the one hand, this can be a strong assumption at times difficult to satisfy in real use-cases. On the other hand, privately finding common variables from multiple Bayesian Networks is a modular operation which can be easily replaced with more advanced techniques, without affecting the core implementation of the `Hybrid CCJT` framework.

Inconsistencies can appear at the level of variable definition (parties using different names for the same concept) or their representation (state/measurement mismatches for discrete and continuous variables, respectively). Some possible mitigation strategies are:

- parties giving synonyms for variable names;
- normalizing variable names (e.g., as lowercase);
- for state alignment mismatches in discrete variables, states not present in all parties could get mapped to a single miscellaneous state;
- for measurement unit mismatches in continuous variables, parties could update their local representations to an agreed-upon unit.

**Consensus on Representation of Common Nodes** Another requirement of `Hybrid CCJT` is that common nodes need to follow the same representation across multiple parties'. For instance, the same discrete variable must have the same set of states when represented by different parties. Common continuous variables must be represented using the same measure unit.

Furthermore, when dealing with a hybrid domain, parties must find a consensus on whether representing a variable as discrete or continuous. On the other hand, when discretizing a continuous variable, we introduce a further consensus problem. That is, variables need to be discretized with the same set of states.

**Modeling limitations of Hybrid CLG BNs** Our work addresses inference in Hybrid CLG BNs leveraging the Lauritzen algorithm [11]. In general, these networks do not allow continuous variables to have discrete children. However, [42] showcase how to extend the Lauritzen algorithm (used in our work) to run exact inference in *augmented Hybrid CLG BNs*. Augmented Hybrid CLG BNs allow continuous variables to have discrete children by modeling CPDs of these nodes as softmax functions. Let $A$ be a discrete node and $Y_1, \ldots, Y_k$ be its (continuous) parents:

$$P[A = a_i] = \frac{\exp(b^i + \sum_{l=1}^{k} w_l^i y_l)}{\sum_{j=1}^{m} \exp(b^j + \sum_{l=1}^{k} w_l^j y_l)} \tag{31}$$

These variables are referred by [42] as CD-CPD (Continuous-Discrete CPD) variables. Briefly, the algorithm proposed requires to:

1. Perform inference as in [11] to compute the strong marginals of $\Delta$ variables and weak marginals of $\Gamma$ variables, without taking into account CD-CPD variables.

2. Insert CD-CPD variables in the Junction Tree, and re-compute marginals of all variables (Referred as *tree re-calibration* in the paper).

Our method allows to perform step 1 of Lerner's algorithm. Extending `Hybrid CCJT` to perform step 2 of Lerner algorithm is an interesting research gap that will be addressed by future work.

