# OpenReview forum: "Collaborative and Confidential Junction Trees for Hybrid Bayesian Networks"
_NeurIPS.cc/2025/Conference — NeurIPS 2025 poster_

### Official Review · Reviewer_6zT8 · 2025-06-18

**Clarity:** 2
**Significance:** 2
**Originality:** 3
**Rating:** 5
**Confidence:** 3

**Summary:**

Bayesian networks are a way to represent distributions over multivariate data, each node is a variable, and a directed edge between nodes shows dependence, the network is a DAG. Collaborative Bayesian networks are when multiple parties determine individual Bayesian networks which may contain some common variables, the goal is to combine networks and perform inference on the combined Bayesian network. However we may also require that agents cannot see each others private local variables.

Junction trees are a standard technique for performing inference over (non-collaborative) Bayesian networks by clustering factors in a graphical model (e.g. a Bayesian network) into nodes called cliques, within each clique one may apply variable elimination followed by message passing across cliques for exact inference, the tree structure ensures that message passing will converge in a single pass.

This paper considers the problem of performing inference over a collaborative Bayesian network while preserving confidentiality of individual parties local variables and their distributions. When there are continuous and discrete variables.

The proposed solution is a custom protocol to build a junction tree that creates two cliques for each agent, one with all agents discrete variables, and one with all the agents continuous variables, and there are two central cliques, one for communal discrete variables and communal continuous variables, and these are connected to ensure that all discrete nodes have no continuous parents.

The method is applied to 9 different datasets synthetically generated from known Bayesian network models and shows improvements over baselines.

**Questions:**

- How does it work with 3+ agents? If I understand correctly, we make a star/hub and spoke network of discrete cliques with interface at the hub and one agent in each spoke, and similar hub+spokes for continuous variables cliques and link the hubs?

- If the interface nodes are the only link from discrete to continuous variables, for an agent with a private discrete parent and private continuous child node, the necessary (private) message must still pass through the interface nodes. How are private threshold variables preserved if they still pass through the interface node? Is it safe to assume that they can be encrypted/ignored by all other agents as they are not required and just pass through? If so, is this equivalent to appending a private continuous clique to each private discrete clique? I understand this preserves Tree structure and strong rootedness.

- What is the “#Arc”  field in the datasets table?

**Ethical Concerns:**

["NO or VERY MINOR ethics concerns only"]

**Final Justification:**

My original comments were mostly minor and my questions were answered accordingly and as expected.

After reading other reviews, particularly rRs9 thoughts on impact and novelty and the issue being addresed, I don't see a clear reason to not accept and have raised my score accordingly.

**Limitations:**

- Cost is exponential in the number of variables in each agent, (presumably one may use nested junction trees?)

- In the two party case, each agent can infer information from the aligned result (which is a fundamental fact of HE and not a problem with Hybrid CCJT), I see this is mentioned in the limitations (Appendix H), this may be more prominently mentioned in the main paper.

**Paper Formatting Concerns:**

I did not notice any formatting errors.

**Quality:**

3

**Strengths And Weaknesses:**

I am familiar with graphical models and junction trees, I am not familiar with all the topics covered in the paper (SMC and I may have misunderstood details and apologize in advance for any mistakes, particularly in .

In general, I am positive about the paper and the approach seems reasonable. My comments are mostly minor.

## Strengths
- Seems intuitive clean simple idea

## Weaknesses
- there is no mention of [Multiply Sections Bayesian Networks](https://ieeexplore.ieee.org/document/858473), presumably one could augment this with HE for interface cliques and perform loopy belief propagation? The continuous variables require a tree structure and strong rooting hence we cannot use such a method?
- the interface cliques must contain shared variables as well as all threshold variables (both private and shared), is there a privacy leak or can we just treat private threshold variables as invisible to irrelevant agents (see questions)?
- it is stated that the two agent case allows each agent to infer the other agents distribution over shared variables, there is a slight data leak?

https://ieeexplore.ieee.org/document/858473
- Inherits the same exponential cost in largest clique as junction trees

---

> ### Author Rebuttal · Authors · 2025-07-31
>
> Thank you for the insightful review.
> Please see our reply to the outlined points.
> We would be happy to clarify any unclear or unaddressed aspects.
>
> ## Weaknesses
>
> **[W1] No mention of Multiply Sectioned Bayesian Networks**
>
> Performing loopy belief propagation over Multiply Sectioned Bayesian Networks would (indeed) break the strong root constraint and lead to poor inference quality in hybrid domains.
> Furthermore, loopy belief propagation entails sending more messages (due to the iterative nature of the algorithm) and is proven to converge only on graphs with at most one cycle [1].
> We will update the related work section with this information.
>
>
> **[W2] Is there a privacy leak over threshold variables?**
>
> Please see the answer to **Q2**.
>
> **[W3] Possible data leak in two-party scenario**
>
> Indeed, when only two parties are involved, the proposed method does not protect the private posterior of interface variables, as doing so is impossible.
> Intuitively, consider computing the sum of $N$ secret numbers $x_1, ..., x_N$ (Equations 7 and 8 in the paper).
> If $N>2$ a certain party $i$ knows $x_i$ and $y = \sum^N_{j = 1}x_j$ from which party $i$ cannot reconstruct the values of $x_{j \neq i}$. If $N=2$, party $i$ can reconstruct $x_j = (x_i + x_j) - x_i = y - x_i$.
> Similar reasoning also holds for products.
>
> The work from Kearns et al. [2] builds upon this basic idea to explore the limitations of confidentiality in Bayesian Networks and specific methodologies including sum-product message passing.
> Notably, that work establishes a generalized definition for confidentiality in Bayesian Network applications (Definition 1 from [2]):
>
> > Let $\Pi$ be any protocol for the $k$ parties to jointly compute the value $y = f(x_1, ... ,x_k)$ from their $n$-bit private inputs. We say that $\Pi$ is privacy-preserving if for every $1 \leq i \leq k$, anything that party $i$ can compute in polynomial time in $n$ following the execution of $\Pi$, they could also compute in polynomial time given only their private input $x_i$ and the value $y$.
>
> We abide by this definition which we include in Appendix B.
>
> ## Questions
>
> **[Q1] How does the protocol work with 3+ parties?**
>
> The junction tree forms 2 star networks corresponding to discrete and continuous variables, respectively (as also remarked by the reviewer), joined via their hubs (i.e., the interface cliques).
> Please note that, as stated in the paper, the Junction Tree dictates the computational logic within the protocol, not the actual distribution of data across parties.
> Thus, an interface clique is not centralized or controlled by any party.
> Instead, parties solve inference collaboratively over this clique.
>
> **[Q2] How are threshold variables protected?**
>
> Unlike interface variables, whose protection depends on homomorphic encryption and secret sharing during alignment and inference, threshold variables are not shared between parties and only used locally.
> We will better emphasize this aspect in the methodology section.
>
> **[Q3] What is the “#Arc” field in the datasets table?**
>
> "#Arc" in Table 5 refers to the number of edges in the original network.
> It describes the number of relationships between variables in the network.
> However, having more edges does not necessarily imply an increase in the number of parameters (#Params column) reported in the same table.
> #Params depends on the network's edge distribution and the representations of variables.
> For instance, a discrete binary variable with two binary parents will require $2^3=8$ parameters. Whereas, one discrete variable with one parent both with three states will require $3^2=9$ parameters.
>
> ## Limitations
>
> **[L1] Cost is exponential in the number of variables in each party**
>
> Per the nature of discrete Bayesian Networks, inference is NP-Hard.
> However, our algorithm is agnostic to the inference algorithm used locally by each party.
> Thus, instead of the Variable Elimination (as employed in our experiments) parties could indeed autonomously harness auxiliary junction trees for local inference exponential in the tree-width instead of the number of variables.
> Note that using nested Junction Trees would not affect communication costs.
>
> **[L2] Undermentioned caveats in two-party case**
>
> As correctly identified by the reviewer, there is an unavoidable confidentiality leak in the two-party case.
> We will update the paper's method section to emphasize the phenomenon.
> More specifically, the issue stems from the need to disclose the output of a bijective function with two inputs, with one of the inputs also being known beforehand.
> Thus, no secure multiparty computation technique, including homomorphic encryption, can address the issue.
> Please refer to the answer of **W3** for additional details.
>
> **References**
>
> [1] Zivan, Roie, Omer Lev, and Rotem Galiki. "Beyond trees: Analysis and convergence of belief propagation in graphs with multiple cycles." AAAI Conference on Artificial Intelligence (AAAI), 2020.
>
> [2] Kearns, Michael, Jinsong Tan, and Jennifer Wortman. "Privacy-preserving belief propagation and sampling." Advances in Neural Information Processing Systems 20 (NeurIPS), 2007.

---

> ### Comment · Reviewer_6zT8 · 2025-08-06
> **Thank  you for the response**
>
> My original comments were mostly minor and my questions were answered accordingly and as expected.
>
> After reading other reviews, particularly rRs9 thoughts on impact and novelty and the issue being addresed, I don't see a clear reason to not accept and have raised my score accordingly.

---

> > ### Author Response · Authors · 2025-08-07
> >
> > We would like to thank the AC and Reviewer 6zT8 for following up.
> > We are glad we could address the reviewer's comments, and are grateful for the positive feedback.

---

### Official Review · Reviewer_HuYN · 2025-07-03

**Clarity:** 3
**Significance:** 3
**Originality:** 4
**Rating:** 4
**Confidence:** 2

**Summary:**

Bayesian Networks (BN) are a tool that allows different parties to collaboratively optimize their production process. State-of-the-art frameworks for collaborative and confidential inference only support discrete data and have high communication costs, while frameworks for hybrid scenarios (mix of discrete and continuous data) are limited to single-party settings. The authors solve this by providing a collaborative and confidential framework for BNs in the hybrid setting, constructing a strongly rooted Junction Tree then running a belief-propagation-like algorithm to combine inference results confidentially. Results show 32% average improvement in predictive accuracy and up to 331× reduction in communication costs.

**Questions:**

Please address weaknesses.

**Ethical Concerns:**

["NO or VERY MINOR ethics concerns only"]

**Final Justification:**

This is definitely a technically strong paper. Authors have addressed my concerns and I am satisfied with their response.
I didn't recommend 5 (my score would be 4.5) because I do not have the domain expertise to judge if this is high-impact and which areas of machine learning this paper can heavily influence (confidence 2). That said, the other reviewers seem to agree that this is a "significant and previously unaddressed challenge".

**Limitations:**

yes

**Quality:**

3

**Strengths And Weaknesses:**

Strengths:
1. The authors' use of Junction Trees gives a clear advantage in terms of communication costs compared to vanilla belief propagation.
2. Authors provide a proof of confidentiality under adversarial model and a clear collaborative strongly-rooted JT protocol.
3. The experimental results are strong: clear improvement in accuracy in hybrid setting, communication cost reduction in discrete setting.

Weaknesses:
1. Could the authors provide a high-level proof sketch for Proof of confidentiality would help understand why the collaborative inference protocol always hides the marginal posteriors on private variables.
2. The Collaborative inference protocol is quite sophisticated and high-level. More implementation details would be appreciated if possible.
3. Limited communication cost improvement analysis in the hybrid setting compared to the extensive discrete results (though Table 1 does show some improvements for hybrid datasets).

---

> ### Author Rebuttal · Authors · 2025-07-31
>
> Thank you for your valuable comments and feedback. It's helpful for us to further improve this work.
> Please see our reply to the highlighted weaknesses.
> We will be happy to follow up on any of the points.
>
> ## Weaknesses
>
> **[W1] No high-level proof of confidentiality**
>
> Below, we outline why both of Hybrid CCJT's collaborative phases preserve confidentiality during inference:
>
> 1. **Collaborative Discrete Inference**
> CPD tables of interface variables (say, $X$) are "augmented" to allocate space for all parents of $X$ $\text{pa}(X)$, some of which might be private. This step is required to ensure the correct outcome of HE, CPD product, and CPD normalization as in Equation 4.
> We show that this step does not leak any information about such private parent variables by proving that their marginals yield a uniform distribution $\mathcal U(\Omega(\text{pa}(X)))$. Where $\Omega(X)$ is the set of states of variable $X$. We prove this in Proposition B.2.
> Then, we prove that, after message passing, the marginal over these parent variables remains uniform. This ensures no information about such private variables leaks at inference time. We prove this in Proposition B.5.
> Furthermore, note that parties encrypt the names of variables and their states to enhance confidentiality further.
> 2. **Collaborative Continuous Inference**
> Unlike discrete variables, continuous counterparts do not require computing and correctly applying normalization factors potentially defined over private factors.
> Thus, continuous private variables are inherently protected during alignment.
> At inference time, the means and variances of shared continuous variables are calculated using multi-party secret sharing, which updates the parameters of such shared continuous interface variables. This approach inherently safeguards the information of private variables, as they are not involved in the update process.
>
> **[W2] Missing implementation details on the inference protocol**
>
> We will prepare a more detailed explanation of the inference protocol in the methodology section and a more detailed pseudocode and a diagram to be included in the appendix.
> The inference is divided in three steps (of which two are collaborative) and follows the logic of the Lauritzen algorithm [1].
>
> 1. **Collaborative Continuous Inference**
> The first step requires marginalizing continuous variables and finding the **strong marginals** over the threshold variables. Recall that threshold variables are the discrete variables with at least one continuous child.
> In short, we aim at computing $P[T|\Gamma = e]$, where $T$ is the set of threshold variables and $\Gamma = e$ is some continuous evidence.
> To do so, parties collaboratively compute the updated parameters of interface continuous variables given the set of continuous evidences as in
> $$
>     {\mu}(X) = \sum_{i \in \mathbb P} w_i\mu_i(X), \quad
>      \sigma(X) = \sum_{i \in \mathbb P} [w_i\sigma_{i}(X)] + \sum_{i \in \mathbb P}[w_i(\mu - \mu_{i}(X))^2
>     ]
> $$
> where the parameters $\mu_i$ and $\sigma_i$ are obtained locally by computing $P[I_\Gamma | \Gamma=e]$.
> Once these new parameters are obtained, they can help compute the marginals over the threshold variables by just marginalizing all the continuous variables.
>
> 2. **Collaborative Discrete Inference**
> Once parties find the marginals of threshold variables, the querying party needs to find the strong marginals over the remaining discrete variables.
> For this, parties compute $P[I_\Delta | \Delta = e]$, the posterior of discrete interface variables given their discrete evidence.
> The querying party receives these posteriors and merge them via secret sharing to find the geometric mean:
> $$
>  P[I_\Delta] = \prod_{i \in \mathbb P} P_i[I_\Delta]^{w_i}
> $$
> The querying party can then use this factor to update its local probability distribution via sum-product message passing [2] (Equation 2 from the paper). For a more detailed explanation on the sum-product message passing please refer to Section 10.2 in the book of Koller [3].
>
> 3. **Local query**
> At this point in the protocol, the querying party can find the posterior of any variable it owns.
> For **discrete variables**, the posterior is found by marginalizing non-target variables, as in a normal junction tree.
> For **continuous variables**, the posterior is found by marginalizing all non-threshold variables and performing weak marginalization:
> $$
>  P[X] = \mathcal N\Bigl(X;
>     \sum_{s \in \Omega(\Delta_{Q})}P[\mathcal T_Q=s] \mu_{s}(X),
>     \sum_{s \in \Omega(\Delta_{Q})}P[\mathcal T_Q=s] \{\sigma_{s}(X) + (\mu - \mu_{s}(X))^2\}
>     \Bigr)
> $$
>
> **[W3] Limited communication cost analysis on hybrid domains**
>
> We repeated the experiments from Table 1 with new variable splits to derive further insights in the communication costs of Hybrid CCJT.
> In Table C isolates the communication costs of Discrete and Continuous collaborative inference. This allows us to better analyze the communication costs gains obtained by natively handling continuous data.
> Furthermore, we report standard deviation measurement of communication costs.
>
> *Table C: Communication cost split for discrete and continuous variables*
> |Dataset||Healthcare||||Sangiovese||||
> |---|---|---|---|---|---|---|---|---|---|
> |#Parties||2|2|4|4|2|2|4|4|
> |Overlap||10%|30%|10%|30%|10%|30%|10%|30%|
> |HybridCCJT|Discrete|$11\pm4.2$|$5\pm4.4$|$20\pm11.6$|$48\pm43.9$|$32\pm0$|$32\pm0$|$96\pm0$|$96\pm0$|
> ||CLG|$0\pm0$|$21\pm4.8$|$0\pm0$|$192\pm62.1$|$29\pm7.5$|$59\pm10.5$|$182\pm44.9$|$541\pm73.6$|
> |$\Delta$-CCJT|3 States|$11\pm3.9$|$21\pm11.2$|$14\pm5.6$|$70\pm24.2$|$21\pm1.0$|$177\pm37.9$|$243\pm45.0$|$3313\pm1251.6$|
> ||5 States|$10\pm3.8$|$20\pm6.7$|$10\pm1.6$|$121\pm78.8$|$38\pm1.1$|$225\pm36.0$|$279\pm46.7$|$35591\pm19277.8$|
> ||10 States|$8\pm3.1$|$39\pm15$|$10\pm1.8$|$229\pm148.7$|$27\pm2.2$|$334\pm146.0$|$539\pm100.2$|$5131\pm1842.4$|
> |CCBNet|3 States|$58\pm27.3$|$61\pm15.6$|$20\pm5.0$|$120\pm30.6$|$240\pm77.1$|$1021\pm298.3$|$25609\pm11412.6$|$28505\pm11896.8$|
> ||5 States|$14.7\pm3.7$|$72\pm29.8$|$10\pm1.6$|$888\pm428.7$|$55\pm7.0$|$1557\pm506.7$|$5210\pm1971.2$|$271053\pm125799.6$|
> ||10 States|$11\pm3.2$|$290\pm139$|$10\pm1.8$|$2068\pm1043$|$169\pm40.8$|$990\pm247.0$|$4766\pm1438.7$|$577838\pm295695.1$|
>
> **References**
>
> [1] Lauritzen, Steffen L., and Frank Jensen. "Stable local computation with conditional Gaussian distributions." Statistics and Computing (SC), 2001.
>
> [2] Shenoy, Prakash P., and Glenn Shafer. "Axioms for probability and belief-function propagation." Machine intelligence and pattern recognition, 1990.
>
> [3] Koller, Daphne, and Nir Friedman. "Probabilistic graphical models: principles and techniques". MIT Press, 2009.

---

> > ### Comment · Reviewer_HuYN · 2025-08-04
> >
> > I appreciate the authors' thorough response. It has addressed most of my concerns.
> > I will keep my score which leans towards acceptance.

---

> > > ### Author Response · Authors · 2025-08-05
> > >
> > > We thank the reviewer for their kind reply and helpful review, which constitutes a valuable source for further improvements to strengthen our work.

---

### Official Review · Reviewer_rRs9 · 2025-07-09

**Clarity:** 4
**Significance:** 3
**Originality:** 4
**Rating:** 5
**Confidence:** 3

**Summary:**

This paper addresses the challenge of performing confidential, collaborative inference in Bayesian Networks (BNs) that span multiple parties and involve hybrid data (i.e., both discrete and continuous variables). The authors identify that existing privacy-preserving methods are typically limited to discrete-only scenarios and often incur high communication overhead.

To overcome these limitations, the paper introduces Hybrid CCJT, a novel framework for secure multi-party inference in hybrid BNs. The proposed method is twofold: first, it presents a novel technique for constructing a single, strongly rooted Junction Tree (JT) across all parties using "interface cliques" to connect their local network structures. Second, it develops a secure protocol based on Multi-Party Computation (MPC) to perform belief propagation on this shared JT, enabling inference while keeping model parameters and structures confidential.

The authors conduct an empirical evaluation of Hybrid CCJT on nine datasets of varying types (mixed, discrete, and continuous). They compare its performance against a state-of-the-art baseline (CCBNet) and an ablation variant of their method ($\Delta$-CCJT), measuring prediction accuracy (using MSE for continuous and Brier score for discrete variables) and communication cost.

**Questions:**

- Could the authors clarify the computational overhead introduced by the cryptographic tools used in the protocol, and how they compare with communication costs in larger models?
- Is the local marginalization in Equation 9 the primary reason for communication efficiency compared to Variable Elimination methods like CCBNet? A brief confirmation of this mechanism would be helpful.
- How does the protocol handle inconsistencies in data representation (e.g., mismatched units or state definitions across parties)?

**Ethical Concerns:**

["NO or VERY MINOR ethics concerns only"]

**Limitations:**

- The paper does not provide a detailed analysis of the computational cost, especially when using cryptographic primitives, which is crucial for practical implementations.
- Data misalignment issues are not sufficiently addressed, and there is no discussion on the protocol's sensitivity to such inconsistencies.
- Scalability concerns with larger datasets or more participants are not fully explored.

**Paper Formatting Concerns:**

NO or VERY MINOR ethics concerns only

**Quality:**

3

**Strengths And Weaknesses:**

* Strengths
The paper addresses a significant and previously unaddressed challenge: confidential inference in collaborative, hybrid Bayesian Networks. It is the first to extend the Junction Tree (JT) algorithm to this domain, enabling the native handling of both discrete and continuous variables without discretization. This greatly enhances the practical relevance of collaborative probabilistic modeling, especially for industrial applications.

The proposed Hybrid CCJT methodology is technically sound, introducing "interface cliques" to construct a strongly-rooted JT in a distributed fashion. This is a critical feature for valid inference in hybrid Conditional Linear Gaussian (CLG) models. The paper’s design is complete, including theoretical proofs and integration of cryptographic primitives to ensure confidentiality.

The empirical evaluation is comprehensive, covering nine datasets with varying characteristics and scales. The comparison against CCBNet and an ablation variant ($\Delta$-CCJT) clearly demonstrates the contribution of the paper’s specific innovations. The reported results show substantial performance improvements, including a 32\% reduction in Mean Squared Error (MSE) and up to a 331x reduction in communication costs, providing strong evidence of the method’s superiority.

* Weaknesses
- Security Guarantee Scope. The security of the protocol is based on a semi-honest, non-colluding adversary model. As noted in Appendix H, the protocol cannot withstand collusion among N-1 participants. While this is a common and explicitly stated assumption, it represents a significant limitation when deployed in environments with low trust or high incentives for collusion. Briefly mentioning this boundary condition in the main text would provide readers with a clearer understanding of the threat model.
- Practical Assumptions for Data Alignment. The successful operation of the framework relies on two key practical assumptions: (a) shared variables between parties can be correctly identified (e.g., through private set intersections on names), and (b) there is consensus on the representation of these shared variables (e.g., the states of discrete variables, applicable units, and discretization schemes). In practice, achieving semantic interoperability is a significant challenge. The importance of this as a prerequisite could be emphasized more clearly.
- Context of the 331x Communication Improvement Claim. The paper reports a communication cost reduction of up to 331x on the Munin #2 dataset (Table 3), which is a highly impressive result. However, this peak performance figure arises under a specific, best-case scenario for the proposed method (a large, purely discrete network where the communication overhead of the baseline is maximal). While the result is technically correct and showcases the potential of the method, highlighting this peak number so prominently (e.g., in the abstract) could be slightly misleading. The average improvement across all datasets and settings might be a more representative measure of the expected gains.

---

> ### Author Rebuttal · Authors · 2025-07-31
>
> Thank you for your positive feedback.
> Please see our reply to the limitations, weaknesses, and questions.
> We look forward to any further comments.
>
> ## Weaknesses
>
> **[W1] Underspecified security guarantee scope in main text**
>
> We will update the abstract and introduction to mention the semi-honest, non-collusion setting.
> Moreover, in the second paragraph of subsection 3.2, we will visually highlight the "semi-honest" denomination alongside the existing one for the "adversarial model".
>
> **[W2] Underemphasized practical assumptions for data alignment in main text**
>
> We will explicitly enumerate the assumptions for successful alignment. Namely, as correctly identified by the reviewer, shared variables across parties should have (i) the same name when defining identical concepts, and (ii) the same discrete states or continuous measurement units.
> Furthermore, we will also point to an appendix with a misalignment-related discussion based on the **Q3** answer below.
>
> **[W3] Undercontextualized $331\times$ communication improvement claim**
>
> We concur that the $331\times$ communication improvement is in a scenario that heavily favours the proposed method.
> We will additionally report the median ($10.4\times$) communication improvement observed alongside the maximum in the abstract and introduction for a fairer presentation.
>
> ## Questions
>
> **[Q1] What is the computational overhead of the cryptographic tools the protocol employs?**
>
> The protocol relies on private set intersection, secret sharing (additive/product-based), and homomorphic encryption (HE) at different points in its lifecycle.
> Among the enumerated cryptographic tools, only HE involves a significant computational overhead, while the primary bottleneck for the others is network latency.
> HE is used exclusively during discrete factor alignment.
> The results below split the HE computation time across:
> 1. *Context Generation*: Generation of HE keys and parameters.
> 1. *Encrypted Column Summation*: The actual HE-encrypted computation; used to compute column normalization factors (i.e., $\alpha$ in Equation 4 in the paper).
> 1. *CPD combination*: Alignment of party CPDs based on the HE-computation outputs.
>
> We report the **per-factor** mean and standard deviation of each HE substep and the total time required for **all factors**.
>
> *Table A: Results on discrete datasets (10% overlap):*
> |Dataset|#Parties|Context Gen. (ms)|Col. Sums (ms)|CPD comb. (ms)|Total (ms)|
> |---|---|---|---|---|---|
> |Asia|2|$153\pm0$|$18.7\pm0$|$0.27\pm0$|176|
> ||4|$644\pm0$|$802\pm0$|$0.53\pm0$|1449|
> ||8|$2155\pm0$|$767\pm0$|$0.746\pm0$|2926|
> |Alarm|2|$136\pm5.7$|$51401\pm82611$|$9.57\pm14.6$|206191|
> ||4|$672\pm50$|$3639\pm4780$|$0.749\pm0.4$|17250|
> ||8|$1900\pm150$|$96745\pm134754$|$4.19\pm4.69$|394601|
> |Child|2|$143\pm5$|$144\pm84$|$0.33\pm0.034$|577|
> ||4|$703\pm26$|$17151\pm9176$|$1.78\pm0.6$|35713|
> ||8|$1984\pm57$|$83010\pm79570$|$3.9\pm2.9$|169999|
> |Link|64|$133\pm7$|$324.9\pm381$|$0.4\pm0.1$|431291|
>
> As expected, substep 2, i.e., encrypted column summation, is the most time-consuming one, taking up to **96** seconds in the worst-case scenario, but remains under one second in 6 out of 10 experiment scenarios.
> Its time requirements increase overall with the number of parties sharing factors.
> In contrast, substeps 1 and 3, i.e., context generation and CPD combination, run in under one second and one millisecond, respectively.
> Moreover, we note that the HE procedure represents a one-off setup cost, after which parties may perform unlimited queries until one of them updates their local network.
>
> *Table B: Results on hybrid datasets (30% overlap):*
>
> |Dataset|#Parties|Context Gen. (ms)|Col. Sums (ms)|CPD comb. (ms)|Total (ms)|
> |---|---|---|---|---|---|
> |Healthcare (Hybrid)|2|$127\pm0$|$26\pm0$|$0.23\pm0$|154|
> |Healthcare (Discrete, 3 states)|2|$142\pm8.24$|$82.4\pm67$|$0.3\pm0.05$|675|
> |Healthcare (Discrete, 5 states)|2|$71.35\pm13$|$71.4\pm62.4$|$0.271\pm0.03$|641|
> |Healthcare (Discrete, 10 states)|2|$139.7\pm10.2$|$165.5\pm152.6$|$0.316\pm0.04$|918|
> |Healthcare (Hybrid)|4|$646.9\pm6.04$|$244\pm90.5$|$0.43\pm0.02$|1785|
> |Healthcare (Discrete, 3 states)|4|$678.46\pm49.3$|$300.2\pm139.2$|$0.48\pm0.06$|2940|
> |Healthcare (Discrete, 5 states)|4|$682.9\pm46.7$|$3007.8\pm3660$|$0.662\pm0.295$|11076|
> |Healthcare (Discrete, 10 states)|4|$677\pm51$|$5956\pm7221.4$|$0.86\pm0.48$|19905|
> |Sangiovese (Hybrid)|2|$127\pm0$|$12.1\pm0$|$0.25\pm0$|141|
> |Sangiovese (Discrete, 3 states)|2|$141\pm9.7$|$827\pm1080$|$0.46\pm0.24$|2906|
> |Sangiovese (Discrete, 5 states)|2|$139.8\pm9.2$|$710.3\pm905$|$0.403\pm0.18$|2553|
> |Sangiovese (Discrete, 10 states)|2|$141\pm11.9$|$621\pm489$|$0.424\pm0.09$|2289|
> |Sangiovese (Hybrid)|4|$127\pm0$|$12.1\pm0$|$0.25\pm0$|140|
> |Sangiovese (Discrete, 3 states)|4|$139\pm9.7$|$813.7\pm1063$|$0.43\pm0.24$|2861|
> |Sangiovese (Discrete, 5 states)|4|$659.3\pm49.6$|$188826.7\pm239278.3$|$13.2\pm16.4$|758015|
> |Sangiovese (Discrete, 10 states)|4|$838\pm51$|$1082339\pm1591371$|$51\pm72.8$|4332927|
>
> When dealing with hybrid domains, we notice that encryption takes significantly more time on the discretized variants.
> While Hybrid CCJT maintains the entire encryption process under one second for both Sangiovese and Healthcare datasets, $\Delta$-CCJT's encryption cost grows as the discretization coarseness is reduced (i.e., more states).
> These results also reaffirm the gains in memory and communication costs of natively handling CLG variables compared to discretizing them, as further highlighted in our answer to **Q2**.
>
> We will add the above tables and discussion as a new appendix.
>
> **[Q2] Is local marginalization in Equation 9 the primary reason for improved communication efficiency?**
>
> Equation 9's local marginalization is the primary reason for improved communication **when only considering discrete variables**.
> When incorporating continuous variables, a significant portion of Hybrid CCJT's efficiency also comes from reducing the representation space from $O(2^N)$ states required by the discretization used in $\Delta$-CCJT or CCBNet, down to $O(N^2)$ parameters required by representing continuous variables in the canonical form.
>
> For more details on the canonical representation of CLG variables, please see Appendix C of the paper.
> Table 1 from the paper contains quantitative results on the improvements given by Equation 9 (i.e., $\Delta$-CCJT vs CCBNet; for example, $15.5\times$ on average for 5 states) and natively handling CLG variables (i.e., Hybrid CCJT vs $\Delta$-CCJT; for example, $1.9\times$ on average for 5 states).
>
> **[Q3] How does the protocol handle data representation inconsistencies?**
>
> While the current protocol assumes no data representation inconsistencies, the name-matching procedure is flexible in being updated to mitigate such issues.
> Inconsistencies can appear at the level of variable definition (parties using different names for the same concept) or their representation (state/measurement mismatches for discrete and continuous variables, respectively).
>
> Some possible mitigation strategies are:
> * parties giving synonyms for variable names;
> * normalizing variable names (e.g., as lowercase);
> * for state alignment mismatches in discrete variables, states not present in all parties could get mapped to a single miscellaneous state;
> * for measurement unit mismatches in continuous variables, parties could update their local representations to an agreed-upon unit.
>
> We will update Appendix H with the above discussion.
>
> ## Limitations
>
> **[L1] No detailed computation time analysis (for cryptographic primitives)**
>
> The computation complexity of homomorphic encryption in solving an overlap is $O(L \cdot N \log(N))$, bounded by $L$ chained multiplications of a size $N$ ciphertext.
> $L$ is the maximum number of parties in an overlap.
> $N$ is a power of two (in the range $2^{10}$--$2^{15}$), derived from the desired precision, the maximum output scale, and $L$.
> Successive multiplications increase complexity linearly [1].
> The cost of a single multiplication is linearithmic [2].
>
> In our implementation, solving a private set intersection between two parties is $O(m \log(m))$ time, where $m$ is the maximum number of variables among the two parties.
>
> For an empirical overview of homomorphic encryption overhead, please refer to our answer to **Q1**.
>
> **[L2] Data misalignment issues insufficiently addressed**
>
> Please see the answer to **Q2**.
>
> **[L3] Scalability concerns not fully explored**
>
> We acknowledge that larger-scale tests on continuous and hybrid networks would be beneficial.
> We note, however, that the primary determiner of our method's scalability compared to a centralized approach is the total number of overlaps and the number of parties involved in each.
> The sizes of each party's local network or the number of parties are not the bottleneck.
> For time complexity, see the answer to **L1**.
> For communication overhead, cost is predominantly related to processing discrete variables.
> As evidenced by our main evaluation results and our reply to **L3** of Reviewer HuYN, continuous variables have much lower communication requirements, making them even less of an issue for scaling.
>
> Moreover, in our motivating example, we expect overlap percentages to be lower than 30% as covered in our experiments.
> Also, in such settings with highly specialized participants, at most a handful of parties would model most concepts.
> Thus, the insights from our experiments with up to 8 parties remain relevant.
> However, we acknowledge that other setups are also plausible (e.g., a central concept modeled by many parties).
>
> **References**
>
> [1] Cheon, Jung Hee, et al. "Homomorphic encryption for arithmetic of approximate numbers." International conference on the theory and application of cryptology and information security (ASIACRYPT), 2017.
>
> [2] Al Badawi, Ahmad, et al. "OpenFHE: Open-source fully homomorphic encryption library." 10th workshop on encrypted computing & applied homomorphic cryptography (WAHC), 2022.

---

> > ### Comment · Area_Chair_f55T · 2025-08-06
> >
> > Reviewer rRs9, the authors have provided extensive details in response to your questions about computational overhead and communication efficiency, among other points raised in your review. How has your opinion of the paper updated since reading their response?

---

### Note · Authors · 2025-08-12

**Dear Reviewers, ACs, and PCs,**

Thank you for your dedication, support and insightful feedback. We deeply appreciate your suggestions, which have greatly enhanced our work. Below is a summary of the key updates and improvements we have made:
* Added ablation studies to assess communication costs of discrete and continuous message passing (Reviewer HuYN)
* Added experiments to assess the computational overhead of cryptographic tools employed by our protocol (Reviewer rRs9)
* Clarified the relationship between communication costs, local marginalization, and discretization (Reviewers rRs9, HuYN)
* Provided high-level proof of confidentiality of the protocol (Reviewer HuYN)
* Discussed details of the inference protocol implementation (Reviewer HuYN)
* Discussed the implementation with multiple agents (Reviewer 6zT8)
* Clarified relationship with previous works such as Multiply-sectioned Bayesian Networks (Reviewer 6zT8)
* Discussed confidentiality in two-party scenarios (Reviewer 6zt8)
* Discussed scalability concerns (Reviewer rRs9)
* Discussed data misalignment requirements and potential solutions to handle inconsistencies (Reviewer rRs9)
* Clarified role of threshold variables in the protocol (Reviewer 6zT8)
* Discussed relationship between inference and hardness of inference in Bayesian Networks (Reviewer 6zT8)

**Best regards,**

The Authors

---

### Decision · Program_Chairs · 2025-09-17

**Decision:**

Accept (poster)

**Comment:**

This paper considers a highly original setting wherein multiple parties (e.g., companies with proprietary data/IP) seek to collaboratively optimize a process (e.g., manufacturing) by performing inference in a large Bayesian network that is composed of separate party-specific Bayesian networks, each representing their respective system or domain. The key challenge in this setting is permitting collaboration while guaranteeing confidentiality---i.e., performing inference in the large Bayesian network while ensuring that parties cannot access the propriety knowledge encoded in other parties' Bayesian (sub-)networks. The paper introduces the "first framework for performing confidential multiparty inference in hybrid domains", where hybrid domains are those involving both discrete and continuous variables.

The reviewers all appreciated the originality and novelty of the setting and the proposed framework. They found the methodology to be sound and compelling, and were satisfied with the empirical results. The following is a representative excerpt from one reviewer's comments:

"The paper addresses a significant and previously unaddressed challenge [...] The proposed methodology Hybrid CCJT is technically sound [...] The paper's design is complete [...] The empirical evaluation is comprehensive."

The authors were highly responsive during the discussion period and largely assuaged the reviewers' concerns, including responding in detail to questions about computational overhead, giving a proof sketch, and providing additional experimental results. The authors also presented a clear plan for revising the paper in response to the reviewers' comments, which I find reasonable and achievable for camera-ready.